# AMPK is a mechano-metabolic sensor linking cell adhesion and mitochondrial dynamics to Myosin-dependent cell migration

Eva Crosas-Molist [1,2], Vittoria Graziani[1,2,13], Oscar Maiques [1,2,13], Pahini Pandya[2], Joanne Monger[1], Remi Samain[1], Samantha L. George[1], Saba Malik[2], Jerrine Salise[2,3], Valle Morales [1], Adrien Le Guennec[2,3], R. Andrew Atkinson [2,3,12], Rosa M. Marti [4], Xavier Matias-Guiu [5,6], Guillaume Charras [7,8], Maria R. Conte [2,3], Alberto Elosegui-Artola [9,10], Mark Holt[2,11] & Victoria Sanz-Moreno [1,2] ✉

Cell migration is crucial for cancer dissemination. We find that AMP-activated protein kinase (AMPK) controls cell migration by acting as an adhesion sensing molecular hub. In 3-dimensional matrices, fast-migrating amoeboid cancer cells exert low adhesion/low traction linked to low ATP/AMP, leading to AMPK activation. In turn, AMPK plays a dual role controlling mitochondrial dynamics and cytoskeletal remodelling. High AMPK activity in low adhering migratory cells, induces mitochondrial fission, resulting in lower oxidative phosphorylation and lower mitochondrial ATP. Concurrently, AMPK inactivates Myosin Phosphatase, increasing Myosin II-dependent amoeboid migration. Reducing adhesion or mitochondrial fusion or activating AMPK induces efficient rounded-amoeboid migration. AMPK inhibition suppresses metastatic potential of amoeboid cancer cells in vivo, while a mitochondrial/AMPK-driven switch is observed in regions of human tumours where amoeboid cells are disseminating. We unveil how mitochondrial dynamics control cell migration and suggest that AMPK is a mechano-metabolic sensor linking energetics and the cytoskeleton.

Cell migration is a key process in immune responses, development, wound healing and tumour dissemination. Cells can migrate within complex 3-dimensional (3D) environments as collective groups or as individual cells, the latter falling mainly into two categories: elongated-mesenchymal and rounded-amoeboid migration. Modes of migration are interchangeable in vitro and in vivo as a consequence of physical and chemical insults[1–4].

Efficient migration requires remodelling of cell-cell contacts and interactions with the extracellular matrix (ECM). The actomyosin cytoskeleton coupled to adhesions form traction points on the ECM to enable cell movement. Elongated-mesenchymal migrating cells rely on integrin-based focal adhesions, while rounded-amoeboid cells are able

to adopt both adhesion-dependent and adhesion-independent migration, where the former employs weak adhesions and the latter involves propulsive, pushing movements[5,6]. Most studies on focal adhesions have been conducted in two-dimensional (2D) environments, lacking the pliability essential for cells to adopt a rounded-amoeboid migration. Yet, mechanical signals in 3D matrices could be very different.

Actomyosin contractility is essential for both elongated-mesenchymal and rounded-amoeboid modes of movement[7,8]. Myosin II-driven contractility relies on multiple kinases. For instance, Rho-kinase (ROCK) or Myosin Light Chain Kinase (MLCK) can regulate Non-muscle Myosin II activity through the phosphorylation of its light

---

and heavy chains[9]. In turn, ROCK can also modulate Myosin activation through inactivating Myosin Phosphatase, leading to increased phosphorylated Myosin Light Chain 2 (pMLC2)[10,11].

On the other hand, cells require energy to migrate, but how metabolism is rewired to fulfil those demands in 3D environments is poorly understood. Mitochondria, considered the powerhouse of the cell, control a number of cellular functions including cell division, apoptosis and cell migration. Mitochondrial activity is intimately linked to their morphology and through mitochondrial dynamics, which include mitochondrial fission and fusion as well as their transport, cells provide high amounts of energy where required[12]. However, the role of mitochondrial dynamics in cells migrating within physiological 3D complex environments employing different modes of movement is poorly defined.

Here, we investigated how cell adhesion to the matrix coupled to ATP levels and mitochondrial dynamics are key to determine the mode of migration that cancer cells adopt in 3D and how this connects to the actomyosin cytoskeleton at the molecular level.

## Results

### Cytoskeleton, adhesion and traction stress in 3D migration

Individual migrating cells can adopt either rounded-amoeboid or elongated-mesenchymal mode of movement. A panel of cell lines was used to define intrinsic patterns of cytoskeletal Myosin II organization and their relationship to cell morphology, adhesion and mode of migration on a pliable 3D matrix. The panel was composed of HT1080 fibrosarcoma, MDA-MB-231 breast cancer and 4 melanoma cell lines: paired WM983A/WM983B, with the same genetic background but derived from primary tumour (WM983A) and lymph node metastasis (WM983B) from the same patient; and paired A375P/A375M2 that share the genetic background since A375M2 were generated from A375P after three rounds of lung metastatic colonization in mice[13-15]. When cultured on a pliable collagen I matrix, within the melanoma group, cells with an elongated morphology showed lower levels of cortical pMLC2 and higher adhesion to the matrix -measured by number of points of attachment to collagen fibres and faster adhesion (Fig. 1a-f and Supplementary Fig. 1a-c). Conversely, amoeboid and metastatic cells displayed a rounded morphology with higher levels of cortical Myosin II activity, membrane blebbing and lower adhesion to the matrix (Fig. 1a-f and Supplementary Fig. 1a-c). The non-melanoma cells HT1080 and MDA-MB-231 adopted an elongated morphology, lower cortical active Myosin II and higher adhesion (Fig. 1a-f and Supplementary Fig. 1a-c). Importantly, active Myosin II had a more diffuse localisation throughout elongated-mesenchymal cells while it was organised as a thick cortex in rounded-amoeboid cells (Fig. 1d). These data show that cell morphology, Myosin II activity/localisation and levels of adhesion are good predictors of the mode of migration (Fig. 1g).

To understand how cells using different modes of migration generate traction stress, 3D image stacks of cells expressing LifeAct-GFP embedded in collagen I matrices were taken over time. For comparison to previous work using beads or artificial manipulation of the matrix[16-18], reflected/backscatter light was collected to image collagen fibres to visualize reorganization induced by traction[19,20]. We then developed a custom-made algorithm for the direct tracking of collagen fibres and direction/angle of displacement in 3D. Integration of strain data, directional information, depth information and stiffness of the matrix allowed the calculation of traction stress (Supplementary Fig. 1d and Supplementary Movies 1-3).

We generated traction maps for maximal intensity projections representing 3D reconstruction of migrating HT1080 and A375M2 cells. While elongated-mesenchymal HT1080 cells exerted traction stresses within the order of 20 Pa, A375M2 rounded-amoeboid cells exerted much lower magnitude stresses (Fig. 1h), consistent with previously reported magnitudes for low adhesion migration[6].

Importantly, inhibition of Myosin II activity with blebbistatin resulted in a reduction of stress exerted by both HT1080 and A375M2 cells (Fig. 1i), revealing that traction stresses applied by migrating cells are Myosin II-dependent.

3D map reconstructions showed that elongated-mesenchymal HT1080 cells exerted traction stresses localised at protrusions (0-2 min) and around the cell body (4-6 min) (Supplementary Fig. 1e), while rounded-amoeboid A375M2 exerted directional stress localised in a single region of the cell (Supplementary Fig. 1f).

In summary, cell morphology, Myosin II organisation and levels of adhesion displayed by cells using different modes of migration determine the magnitude of the traction stresses exerted into the matrix (Fig. 1g).

### Different modes of migration and mitochondrial metabolism

How cells obtain the energy required for generating traction stress in 3D systems remains poorly understood. NMR metabolomics on rounded-amoeboid WM983B and A375M2 cells compared with their elongated-mesenchymal counterparts, WM983A and A375P, respectively, revealed that the metabolic profiles of each pair of cells were statistically different using partial least squares discriminant analyses (PLS-DA) (Supplementary Fig. 2a). To understand better their differential metabolism, we analysed engagement in glycolysis vs mitochondrial respiration. The extracellular acidification rate (ECAR) of the media showed that aerobic glycolysis was not specifically associated with either type of migration (Supplementary Fig. 2b, c) or any specific oncogene/genetic alteration (Supplementary Fig. 2d). In contrast, higher mitochondrial respiration was associated with elongated-mesenchymal migration (Fig. 2a, b and Supplementary Fig. 2e-h). As such, higher oxygen consumption rates (OCR) and ATP production derived from oxidative phosphorylation (OXPHOS) were detected in elongated-mesenchymal cells when compared with rounded-amoeboid moving cells, with greater differences when cells were embedded in a pliable collagen matrix, where cells can more readily adopt different modes of migration (Fig. 2a, b and Supplementary Fig. 2e-h). In 3D, the biosensor PercevalHR[21] revealed higher ATP/ADP ratios in elongated-mesenchymal cells (Fig. 2c, d and Supplementary Fig. 2i). On the other hand, AMP levels were higher in rounded-amoeboid WM983B and A375M2 cells compared to their elongated-mesenchymal pairs (Fig. 2e). Collectively, these data suggest that cells exerting strong traction stresses require higher levels of energy from oxidative phosphorylation while cells adopting a rounded-amoeboid migration engage less avidly in mitochondrial respiration.

We have shown that the magnitude of the stress that cells exert on the matrix relates to the adhesion levels to that matrix (Fig. 1e-h). Considering the data shown above, we hypothesised that levels of adhesion could control the energy demands required for different migratory modes. The analysis of ATP levels comparing adherent cells versus floating cells revealed that when cells are challenged with a non-adherent environment, similar to what we observed in weakly adhesive rounded-amoeboid cells, they decreased ATP levels, particularly ATP derived from mitochondrial respiration (Fig. 2f, g). Accordingly, we observed lower mitochondrial membrane potential in cells under floating conditions (Fig. 2h and Supplementary Fig. 2j), in correlation with low OXPHOS in weakly-adherent rounded-amoeboid cells (Fig. 2a, b). Cell viability was not compromised under non-adherent conditions (Supplementary Fig. 2k). Altogether, these data suggest that low adhesion in migrating cells could influence their energetic demands.

### DDR1 controls adhesion levels and energy demands

We next hypothesized that adhesion receptor levels in cells could have an impact in mode of migration and energy requirements. Most of the migratory surfaces including collagens, laminins and fibronectin engage with integrin receptors[22] while Discoidin Domain Receptors (DDRs) also participate in cell adhesion to fibrillar collagens[23]. Using

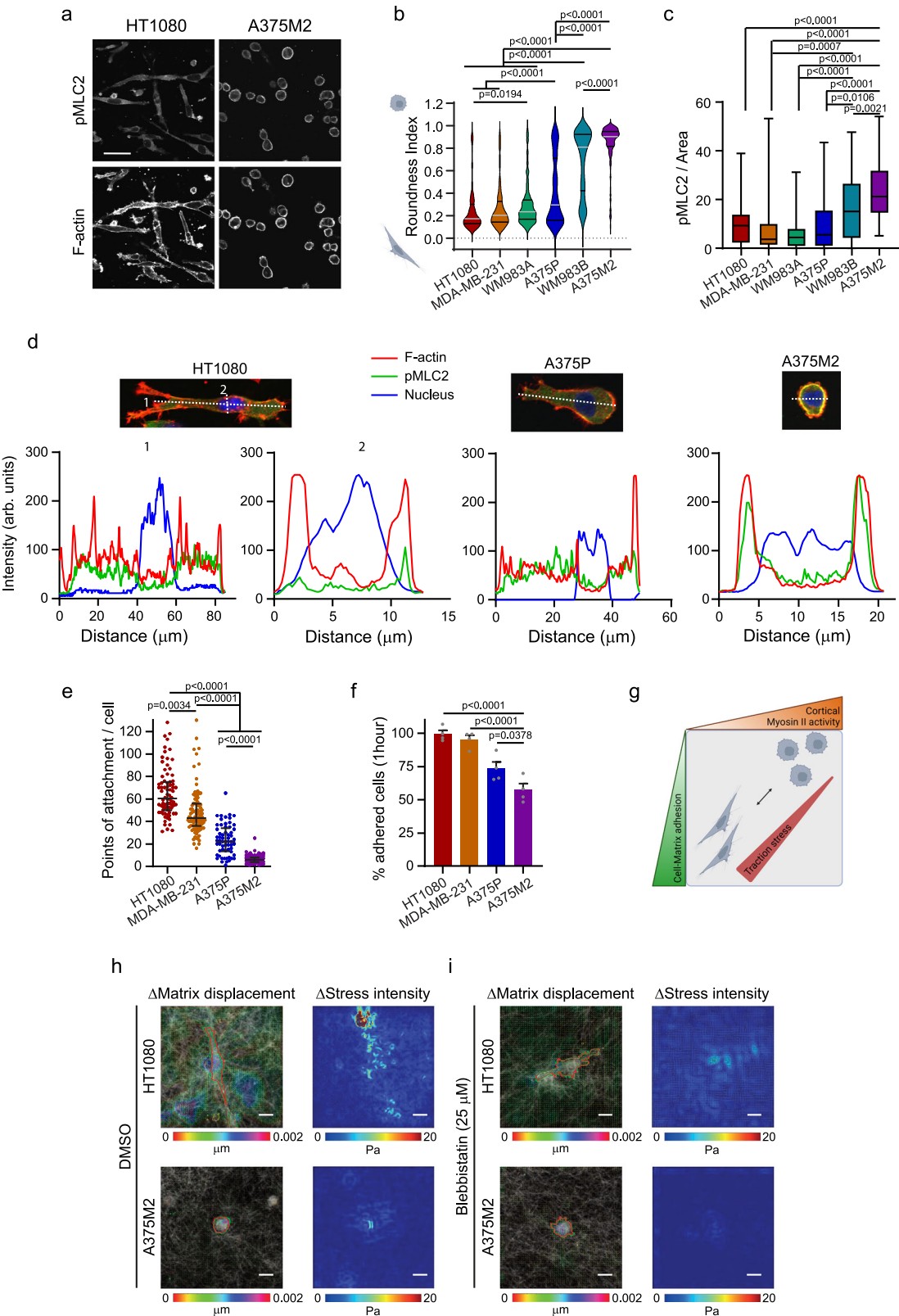

mRNA data from Affymetrix microarrays comparing A375M2 with A375P cells on a collagen I matrix or A375M2 cells with lowered Myosin II activity[4], we observed that rounded-amoeboid cells expressed good levels of collagen I binding integrins (Fig. 3a). However, DDR1 mRNA levels were low in these cells (Fig. 3a, b). Importantly, DDR1 protein expression directly correlated with adhesion to the collagen I matrix (Fig. 3c) and inversely correlated with cytoskeletal features of

amoeboid-ness (Fig. 1). Silencing DDR1 in elongated-mesenchymal WM983A and A375P cell lines decreased adhesion to collagen I matrix (Fig. 3d), and induced amoeboid traits: led to increased cell rounding and Myosin II activation, increased membrane blebbing and increased 3D invasion (Fig. 3e–h), without affecting cell viability (Supplementary Fig. 3a). Similar results were observed using three independent on-target siRNA sequences in both cell lines (Supplementary Fig. 3b–e).

**Fig. 1 | Cytoskeleton, adhesion and traction stress in 3D migration.** Cells seeded on a 3D collagen I matrix. **a** pMLC2 and F-actin confocal images in HT1080 and A375M2 cells ($n = 3$). Scale bar = 50 μm. **b, c** Quantification of cell morphology (211, 291, 313, 358, 339 and 311 cells pooled from $n = 6$) (**b**) and pMLC2 immuno-fluorescence signal normalized by cell area (45, 72, 65, 79, 85 and 83 pooled from $n = 3$) (**c**). **d** Representative fluorescence intensity line scans (dashed white lines in image) showing distribution of F-actin (red), pMLC2 (green) and nucleus (blue) along elongated-mesenchymal (HT1080 and A375P) and rounded-amoeboid (A375M2) cells ($n = 10$ cells/cell line). **e** Quantification of points of attachment between cells and matrix (78, 99, 61 and 112 cells pooled from $n = 3$). **f** Percentage of adhered cells 1 h after seeding on a matrix of collagen I ($n = 4$). **g** Scheme showing that cell morphology, Myosin II activity and adhesion to the matrix define the mode of migration and the stress exerted into the matrix (Created with BioRender.com). **h, i** Cells treated with vehicle (DMSO) or blebbistatin (25 μM). (Left) Representative

displacement vector maps from maximum intensity projections of cells obtained from the tracked displacements of collagen I fibres between 0 to 2 min. Red and green boundaries indicate the outline of the cell in the previous and current frames, respectively. Scale bar = 20 μm. (Right) Representative traction stress magnitude maps corresponding to displacement maps. Colour bar indicates traction stress magnitude (Pascal, Pa). Scale bar = 20 μm. Dot plot (**e**) shows median with inter-quartile range (each dot represents a single cell). Violin plot (**b**) shows median with interquartile range. Box plots (**c**) show median (centre line), interquartile range (box) and min-max values (whiskers). Graph (**f**) shows mean ± SEM. *p* values by Kruskal–Wallis test with Dunn's multiple comparisons test (**b, c, e**), one-way ANOVA with Dunnett's correction versus A375M2 (**f**). All *n* are indicative of independent experiments unless otherwise stated. Source data are provided as a Source Data file. See also Supplementary Fig. 1.

On the other hand, HT1080 and MDA-MB-231 cells exhibited higher levels of β1 integrins compared to A375M2 cells (Supplementary Fig. 4a, b), in accordance with faster adhesion and higher number of points of attachment to the matrix (Fig. 1e, f). Reducing levels of *ITGB1* (β1 integrin subunit) in highly adhesive and elongated cells increased cell roundness, while having no effect on viability (Supplementary Fig. 4c–e, g). Nevertheless, contrary to the observed rounding response upon DDR1 depletion, roundness upon *ITGB1* silencing was not accompanied by any increase in Myosin II activity or membrane blebbing (Supplementary Fig. 4d, f). Indeed, data from the Affymetrix microarray showed higher expression levels of *ITGB1* in A375M2 cells compared to A375P cells or A375M2 cells treated with contractility inhibitors (Fig. 3a). In agreement, rounded-amoeboid WM983B and A375M2 cells expressed higher protein levels of β1 integrins compared to their elongated counterparts WM983A and A375P (Supplementary Fig. 4h). These data suggest that β1 integrins may be required for both modes of migration in 3D.

Overall, our results indicate that elongated-mesenchymal cells require actomyosin contractility coupled to stronger adhesions to migrate, and define DDR1 as a key collagen I receptor for elongated-mesenchymal movement that is dispensable for rounded-amoeboid migration.

Importantly, DDR1 silencing in elongated-mesenchymal cells cultured on a collagen I matrix resulted in decreased OXPHOS and lower ATP levels (Fig. 3i, j), characteristic of rounded-amoeboid migrating cells (Fig. 2), further confirming the crosstalk between adhesion to the matrix and energy demands.

## ATP levels control plasticity of cell migration through AMPK

Next, we investigated how different energy requirements could result in cytoskeletal remodelling. Rounded-amoeboid cells harbour lower ATP/ADP ratio and higher AMP levels compared to elongated-mesenchymal cells (Fig. 2d, e). An imbalance in the ATP/ADP or ATP/AMP ratio leads to phosphorylation and activation of AMPK[24–26], the main energy sensor in cells. On the other hand, Myosin II activity is regulated by the phosphorylation of MLC and by Myosin Phosphatase inactivation[9–11]. A chemical genetic screen identified MYPT1 and MBS85, regulatory subunits of the Myosin Phosphatase complex, as new direct substrates of AMPK[27]. Phosphorylation of MYPT1 and MBS85 at Ser472 and Ser452 respectively inhibits Myosin Phosphatase activity by preventing the binding of MYPT to Myosin II, leading to Myosin II activation[27,28]. Perturbing mitochondrial respiration with rotenone and antimycin A[29] induced AMP accumulation and a decrease in ATP/AMP and ATP/ADP ratios that resulted in AMPK activation (as measured by pThr172-AMPK and pSer79-ACC) (Fig. 4a, b and Supplementary Fig. 5a). We hypothesized that low mitochondrial metabolism in weakly adherent cells results in an ATP/AMP or ATP/ADP imbalance that leads to AMPK activation and subsequent MYPT1 inactivation, MLC2 phosphorylation and rounded-amoeboid migration. We first compared AMPK phosphorylation levels and found that

rounded-amoeboid cells harbour higher intrinsic levels of pAMPK compared to their elongated-mesenchymal counterparts (Fig. 4c and Supplementary Fig. 5b). Interestingly, AMPK activation was accompanied by higher levels of phosphorylated, and therefore inactivated, MYPT1 (as measured by pSer472-MYPT1) (Fig. 4c and Supplementary Fig. 5b), in agreement with higher levels of Myosin II activation at the cortex of rounded-amoeboid cells (Fig. 1c, d). We next knocked-down AMPK (*PRKAA1* and *PRKAA2*) using RNAi in A375M2 cells resulting decreased Ser472-MYPT1 phosphorylation (Fig. 4d). This phenomenon coincided with loss of amoeboid features, decreased cell rounding, reduced Myosin II activity and increased traction stresses (Fig. 4e–h). An increase in the XF ATP rate index (ratio between mitochondrial and glycolytic ATP) was also observed (Supplementary Fig. 5h). Similar results were obtained when treating WM983B and A375M2 rounded-amoeboid cells with the AMPK inhibitor Compound C[27], decreased MYPT1 phosphorylation, loss of cell roundness and reduced Myosin II activity (Supplementary Fig. 5c–g). Conversely, treatment of A375P and WM983A elongated-mesenchymal cells with the AMPK activator A769662[27], led to increased pAMPK and pMYPT1 levels (Fig. 4i and Supplementary Fig. 5i, j), resulting in the induction of rounded-amoeboid behaviour (Fig. 4j–l and Supplementary Fig. 5k–m). Strikingly, transition towards an amoeboid behaviour induced by silencing of DDR1 in elongated-mesenchymal cells was associated with increased AMPK activity and MYPT1 phosphorylation (Fig. 4m, n) and was prevented by AMPK inhibition (Fig. 4m–q).

Overall, our data suggest that DDR1-driven adhesion/migration in elongated-mesenchymal cells requires high levels of ATP. By lowering DDR1-dependent adhesion, mitochondrial respiration is decreased and AMPK activated, inducing a transition towards rounded-amoeboid migration.

## Mitochondrial dynamics in 3D migration

By remodelling the mitochondrial network, cells provide energy in specific subcellular regions with high demands. Through mitochondrial dynamics, which includes processes of mitochondrial fission and fusion as well as their transport, cells regulate the size, number, morphology and localization of mitochondria[12]. AMPK has been associated to mitochondrial dynamics through the direct phosphorylation of Mitochondrial Fission Factor (MFF)[29]. MFF is one of the main receptors that recruit and stabilize Dynamin-related protein 1 (DRP1) to the outer mitochondrial membrane, allowing mitochondrial fission to be carried out[30]. We corroborated that activation of AMPK using A769662 in elongated-mesenchymal cells resulted in increased phosphorylation of MFF and increased mitochondrial fission, as measured by reducing mitochondrial branching (Fig. 5a, b and Supplementary Fig. 6a, b), while knocking-down AMPK (siPRKAA1/2) in rounded-amoeboid cells resulted in lower MFF phosphorylation levels and fused mitochondria (Fig. 5c, d and Supplementary Fig. 6c). These data show that AMPK activity levels are connected to both mitochondrial and cytoskeletal dynamics. We, therefore, decided to study the relevance of

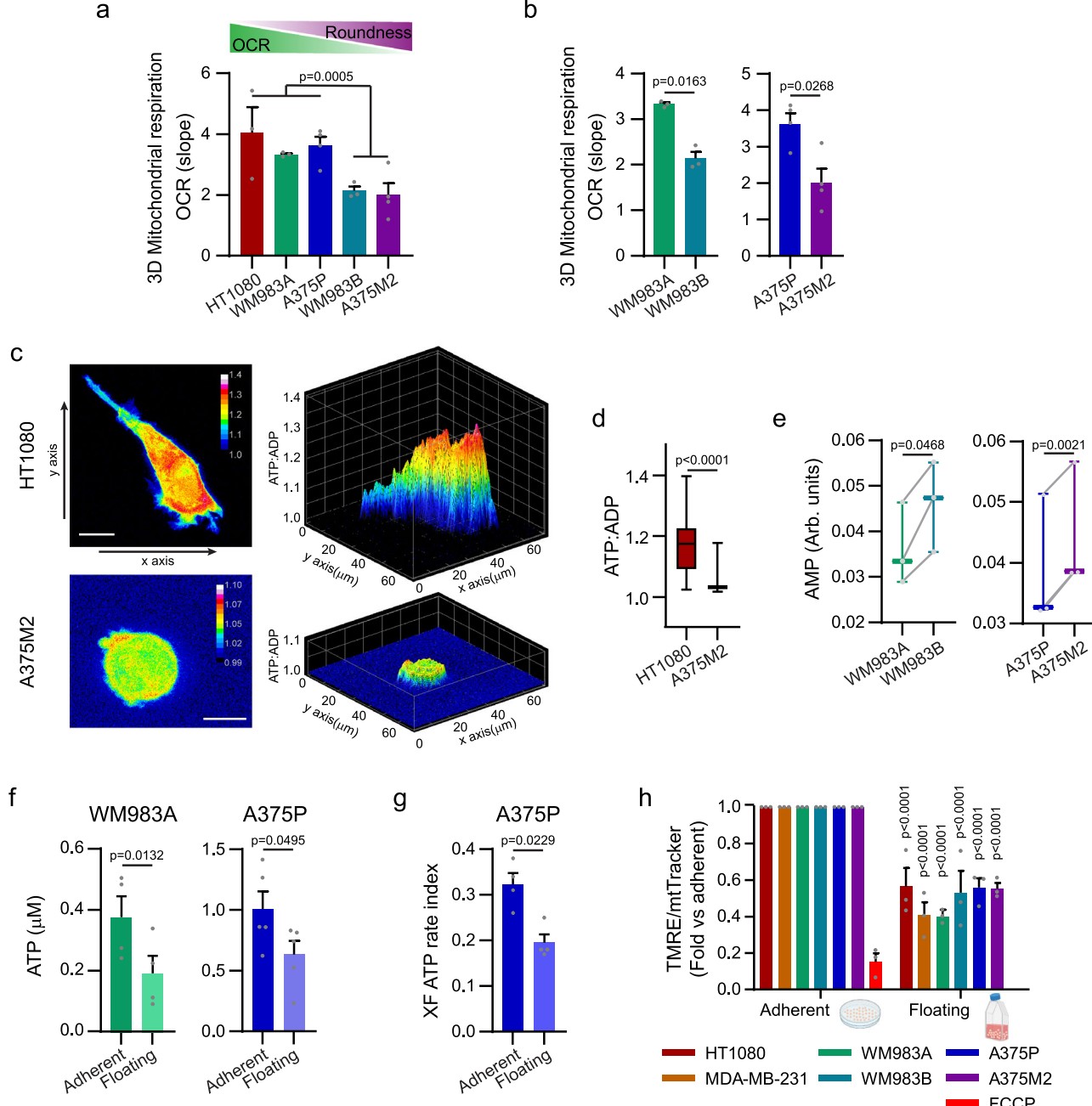

**Fig. 2 | Different modes of migration and mitochondrial metabolism.**
**a**, **b** Oxygen consumption rate (OCR) from cells embedded in a 3D collagen I matrix (*n* = 3). Comparison between elongated-mesenchymal and rounded-amoeboid cells in a panel of cell lines and between the melanoma pairs. **c** Maximal intensity projection (left) and 3D representation (right) of ATP:ADP ratio in HT1080 and A375M2 cells expressing Perceval HR biosensor, embedded in a 3D collagen I matrix. Scale bar = 10 µm. **d** Quantification of ATP:ADP ratio from (**c**) (18 cells/condition pooled from *n* = 3). **e** AMP levels quantified by ¹H-NMR for the indicated cell lines (*n* = 3). **f** Intracellular ATP levels in WM983A and A375P cells grown under adherent or floating conditions for 24 h (WM983A, *n* = 4; A375P, *n* = 5). **g** XF ATP rate index indicative of the ratio of mitochondrial and glycolytic ATP Production Rate in A375P under adherent or floating conditions for 24 h (*n* = 4). **h** Quantification of

Tetramethylrhodamine Ethyl Ester Perchlorate (TMRE) fluorescence intensity normalized by MitoTracker Deep Red signal by FACS in cells adhered on plastic or floating cells for 24 h. Data shown as fold versus adherent cells (*n* = 3). Graphs (**a**, **b**, **f**, **g**, **h**) show mean ± SEM. Box plots (**d**, **e**) show median (centre line), inter-quartile range (box) and min-max values (whiskers). *p* values were calculated using two-tailed tests (**a**, **b**, **d**–**g**). *p* value by unpaired *t*-test comparing elongated-mesenchymal versus rounded-amoeboid cells (**a**), unpaired *t*-test (**b**), paired *t*-test (**e**–**g**), Mann-Whitney test (**d**) and two-way ANOVA with Sidak's correction comparing floating vs adherent cells (**h**). All *n* are indicative of independent experiments unless otherwise stated. Source data are provided as a Source Data file. See also Supplementary Fig. 2.

mitochondrial dynamics in regulating the cytoskeleton and individual cell migration.

We compared mitochondrial organization in the panel of cell lines growing on a collagen I matrix. Mitochondria of rounded-amoeboid cells (WM983B and A375M2) were punctate and

disconnected while mitochondria in their elongated-mesenchymal counterparts (WM983A and A375P), and in HT1080 and MDA-MB-231 cells, were long and formed a network (Fig. 5e, Supplementary Fig. 6d–f and Supplementary Movies 4, 5). Elongated mitochondria from mesenchymal cells harboured higher mitochondrial

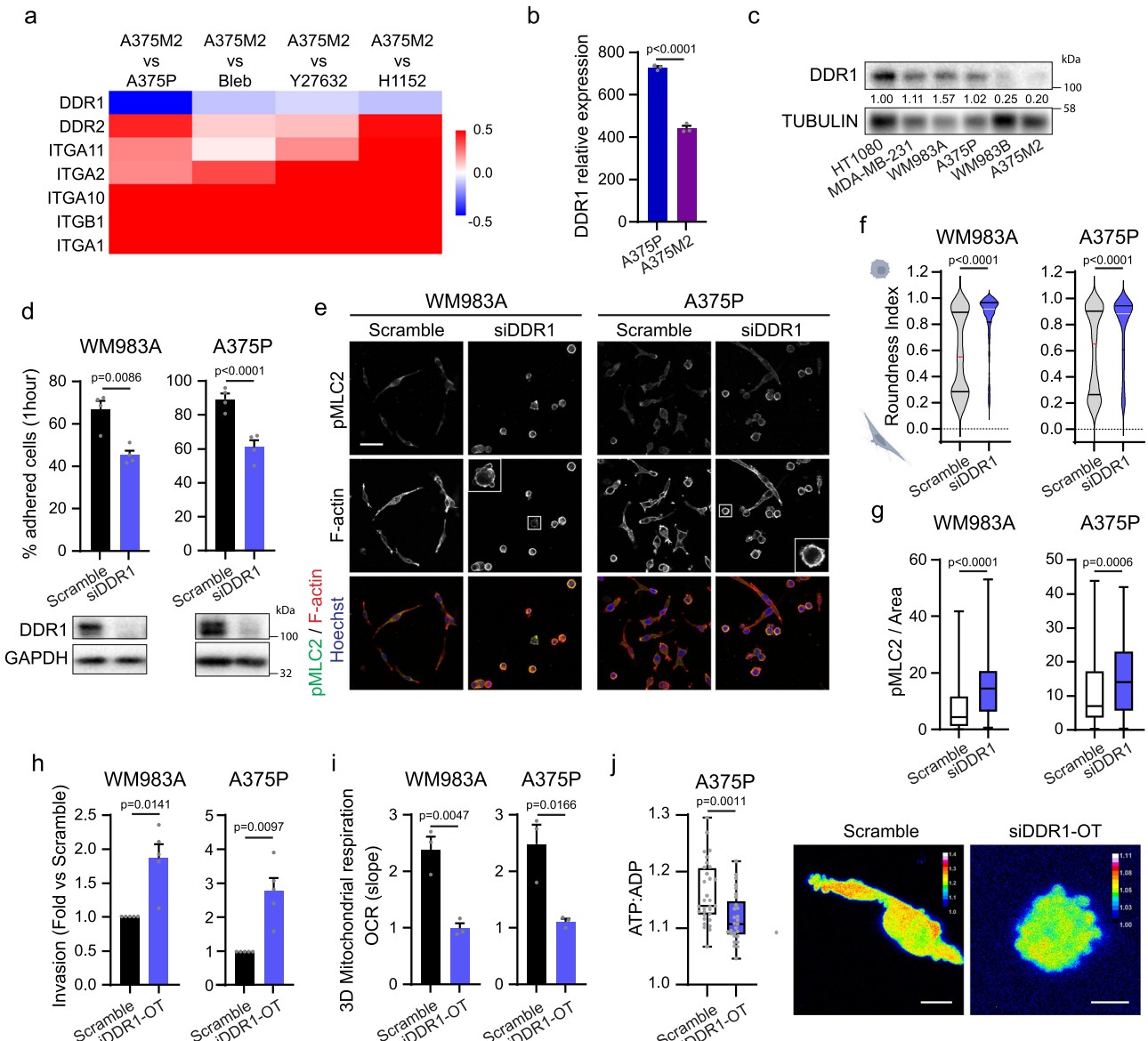

**Fig. 3 | DDR1 controls adhesion levels and energy demands. a** Heat map showing Log$_2$ data from an Affymetrix microarray comparing A375M2 versus A375P cells or A375M2 cells seeded on collagen I and treated with Blebbistatin (Bleb) or ROCK inhibitors (Y27632 or H1152) for 24 h (GSE23764). **b** Expression levels of DDR1 from an Affymetrix microarray comparing A375P and A375M2 cells ($n = 3$). **c** DDR1 levels in a panel of cell lines ($n = 5$). Quantification normalized by TUBULIN. **d** (Top) Quantification of adhered cells after 1 h seeding on a collagen I matrix ($n = 4$). (Bottom) DDR1 protein levels after DDR1 knock-down. **e** Cells seeded on a collagen I matrix. Immunofluorescence images showing pMLC2 levels (green), F-actin (red) and Hoechst (blue) ($n = 3$). Scale bar = 50 μm. Inset shows blebbing cell for siDDR1. **f** Quantification of cell morphology ($n = 3$). **g** Quantification of pMLC2 immuno-fluorescence signal normalized by cell area (107, 105, 116, 115 cells pooled from

$n = 3$). **h** 3D invasion index into a collagen I matrix ($n = 5$). **i** OCR from cells embedded in a 3D collagen I matrix ($n = 3$). **j** Quantification (left) and maximal intensity projection (right) of ATP:ADP ratio in A375P cells expressing Perceval HR biosensor, embedded in a 3D collagen I matrix (31, 29 cells pooled from $n = 3$). Scale bar = 10 μm. Graphs (**b**, **d**, **h**, **i**) show mean ± SEM. Violin plots (**f**) show median with interquartile range (each dot represents a single cell). Box plots (**g**, **j**) show median (centre line), interquartile range (box) and min-max values (whiskers) (each dot in (**j**) represents a single cell). $p$ values were calculated using two-tailed tests (**b**, **d**, **f**–**j**). $p$ value by unpaired $t$-test (**b**, **d**, **i**, **j**), one sample $t$-test (**h**) and Mann–Whitney test (**f**, **g**). All $n$ are indicative of independent experiments unless otherwise stated. Source data are provided as a Source Data file. See also Supplementary Fig. 3 and 4.

membrane potential than fragmented mitochondria from rounded-amoeboid cells (Fig. 5f and Supplementary Fig. 6g). These data together with higher OXPHOS activity detected in elongated-mesenchymal cells (Fig. 2a, b), suggested that mitochondria in these cells are more active. Strikingly, silencing DDR1 in A375P cells resulted in spherical disconnected mitochondria, which was prevented by inhibiting AMPK (Fig. 5g). These data suggest a crosstalk between adhesion, energy levels and mitochondrial dynamics. Overall, mitochondrial fusion and metabolism were higher in elongated-mesenchymal moving cells, possibly to fulfil the

energetic demands required for higher levels of adhesion and traction stress.

**Mitochondrial dynamics control the cytoskeleton and invasion**

We further analysed the relevance of mitochondrial dynamics for cell migratory plasticity. Mitochondrial morphology is modulated by expression levels, activity and localization of mitochondrial fission and fusion proteins. AMPK activation not only induces a transition to a rounded-amoeboid phenotype but also results in MFF phosphoryla-tion, inducing mitochondrial fission (Fig. 5a, b and Supplementary

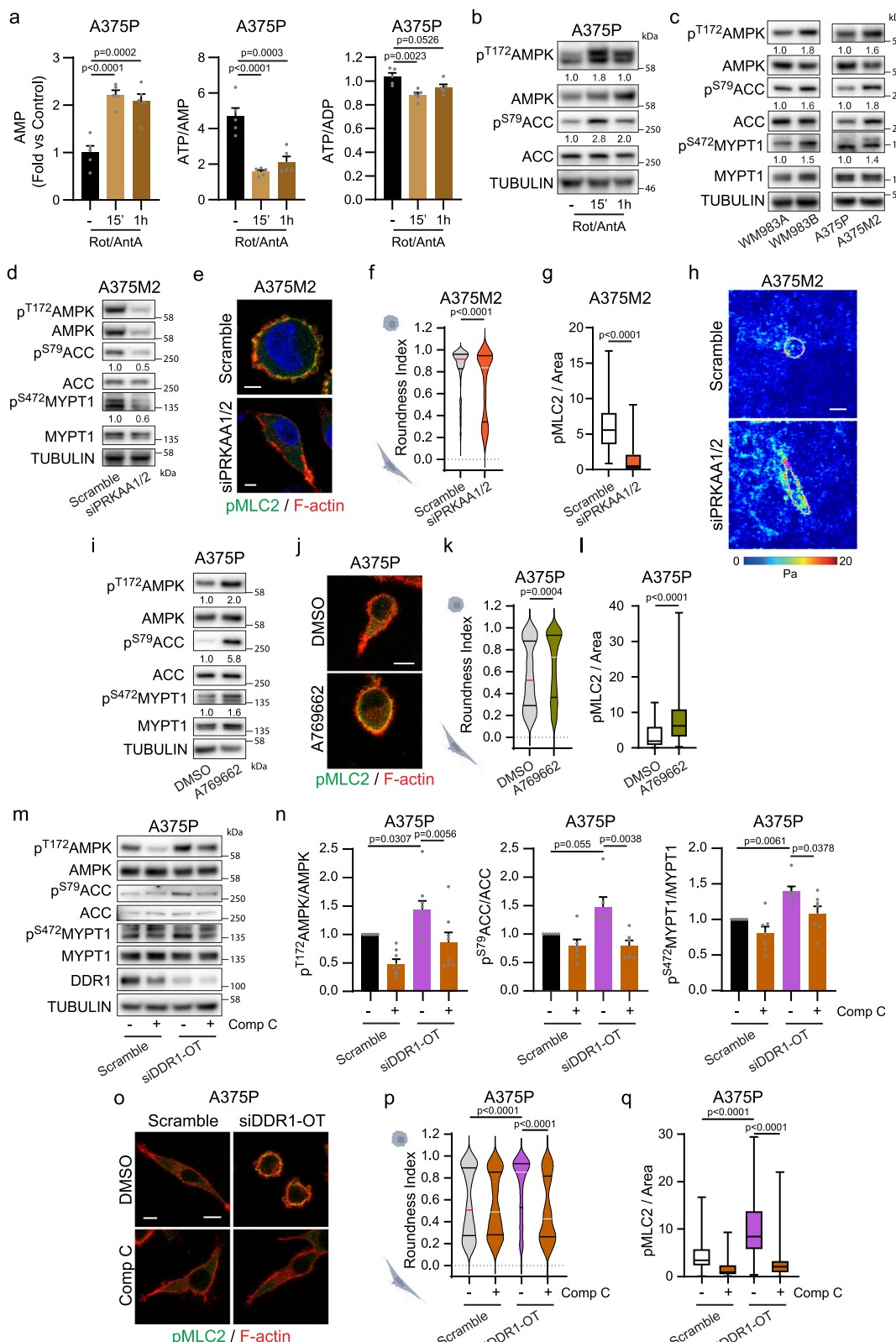

Fig. 6a, b). Interestingly, preventing mitochondrial fission via siMFF or siDNM1L in rounded-amoeboid cells resulted in elongated mitochondria (Fig. 6a, b). This, in turn, increased mitochondrial respiration and ATP production, decreased AMPK activity and MYPT1 phosphorylation (Fig. 6a and Supplementary Fig. 7a–c), leading to lower Myosin activity, loss of amoeboid features and decreased invasion into the matrix (Fig. 6c–f).

In addition, we found that elongated-mesenchymal WM983A and A375P cells expressed higher levels of mitochondrial fusion proteins when compared to their rounded-amoeboid counterparts WM983B and A375M2 (Fig. 7a), in accordance with their mitochondrial morphology (Fig. 5e), which could be a long term adaptation to their energy demands. During mitochondrial fusion, mitofusin 1 and 2 (MFN1 and MFN2) form homo- and hetero-oligomeric complexes to

**Fig. 4 | ATP levels control plasticity of cell migration through AMPK. a** AMP, ATP/AMP and ATP/ADP in A375P cells treated with rotenone (0.5 μM) and antimycin A (0.5 μM) (5 replicates/condition). **b** Western blot for the same conditions as in (**a**). **c** Western blot of the indicated proteins (WM983 $n = 6$, A375 $n = 5$). **d** Western blot upon AMPK knock-down ($n = 3$). **e** Cells grown on a collagen I matrix. Immunofluorescence images showing pMLC2 (green) and F-actin (red) ($n = 3$). Scale bar = 5 μm. **f–h** Quantification of cell morphology (211, 193 cells pooled from $n = 3$) (**f**), pMLC2 immunofluorescence signal normalized by cell area (52, 66 cells pooled from $n = 3$) (**g**) and representative traction stress ($n = 3$). Colour bar indicates traction stress magnitude (Pascal, Pa), scale bar = 20 μm (**h**). **i** Western blot upon AMPK activation (A769662 10 μM, 30 minutes) ($n = 5$). **j–l** Cells grown on a collagen I matrix. Immunofluorescence images showing pMLC2 (green) and F-actin (red) after A769662 treatment (10 μM, 24 h) ($n = 3$), scale bar = 10 μm (**j**); quantification of cell morphology (317, 283 cells pooled from $n = 3$) (**k**) and quantification of pMLC2 immunofluorescence signal normalized by cell area (81, 84 cells pooled from $n = 3$) (**l**). **m, n** Western blot and quantification after DDR1 knock-down and Comp C treatment (2 μM, 24 h) ($n = 8$ AMPK, $n = 6$ ACC, $n = 9$ MYPT1). **o–q** Cells grown on a collagen I matrix. Immunofluorescence images showing pMLC2 (green) and F-actin (red) ($n = 3$), scale bar = 10 μm (**o**); quantification of cell morphology (274, 210, 274, 191 cells pooled from $n = 3$) (**p**) and quantification of pMLC2 immunofluorescence signal normalized by cell area (93, 84, 94, 93 cells pooled from $n = 3$) (**q**). Quantification versus corresponding total protein (**b, c, d, i**). Graphs (**a, n**) show mean ± SEM. Violin plots (**f, k, p**) show median with interquartile range. Box plots (**g, l, q**) show median (centre line), interquartile range (box) and min-max values (whiskers). $p$ values were calculated using two-tailed tests (**f, g, k, l**). $p$ value by Mann–Whitney test (**f, g, k, l**), one-way ANOVA with Dunnett's correction (**a**) or Tukey's multiple test (**n**) and Kruskal-Wallis with Dunn's multiple comparisons test (**p, q**). All $n$ are indicative of independent experiments unless otherwise stated. Source data are provided as a Source Data file. See also Supplementary Fig. 5.

tether and fuse the outer membranes of two mitochondria, while OPA1 Mitochondrial Dynamin Like GTPase (OPA1) directs mitochondrial inner membrane fusion[31]. Down-regulation of MFN1 and MFN2 in elongated-mesenchymal WM983A and A375P cells, not only altered their mitochondrial network to spherical disconnected mitochondria, decreasing mitochondrial respiration and ATP (Fig. 7b, d and Supplementary Fig. 7d–f), but it also increased cell rounding, cortical Myosin II activity and membrane blebbing (Fig. 7e–g and Supplementary Fig. 7g–i). This switch resulted in decreased adhesion to collagen I, increased 3D invasion, and decreased traction stress (Fig. 7h–j and Supplementary Fig. 7j, k). These data were further validated using four alternative on-target siRNA sequences against MFN1 in both elongated-mesenchymal cell lines (Supplementary Fig. 7l–o). Similar results were obtained via knocking down OPA1 (regulates mitochondrial inner membrane fusion) in both WM983A and A375P cells (Supplementary Fig. 8a–g). Cell viability was not compromised after any siRNA transfection (Supplementary Fig. 7p).

Importantly, rounded-amoeboid behaviour induced by preventing mitochondrial fusion was mediated by increased AMPK and MYPT1 phosphorylation, and it was prevented by AMPK inhibition using Compound C (Fig. 7k–o and Supplementary Fig. 8h–j).

Altogether, these data indicate that mitochondrial dynamics downstream of cell adhesion play a crucial role in the crosstalk between AMPK and cytoskeletal remodelling in migrating cells.

### Adhesion, AMPK and mitochondrial dynamics in vivo

Tumours derived from WM983B and A375M2 cells have higher levels of Myosin II activity (Fig. 8a, b) and cell roundness compared to tumours from WM983A and A375P cells, respectively[14], indicating that our in vitro 3D systems replicate cell behaviour in vivo. Accordingly, lower levels of DDR1, MFN2 and OPA1 and higher levels of pAMPK were detected in WM983B and A375M2 tumours, compared to WM983A and A375P tumours (Fig. 8a, b) as shown in vitro (Figs. 3c and 7a).

We have previously shown that cancer cells at the invasive front (IF) of tumours have amoeboid features: they are mainly rounded and harbour higher levels of active Myosin II when compared with cells within the tumour body (TB)[4,14,20,32,33]. Accordingly, melanoma cells at the IF of human primary tumours showed lower levels of DDR1, MFN2 and OPA1, while harbouring higher levels of pAMPK and higher amoeboid score (Fig. 8c, d and Supplementary Fig. 9a). Data showed a positive correlation between mitochondrial fusion proteins and DDR1 levels (Fig. 8e and Supplementary Fig. 9b). These data suggest that cells in the invasive areas of the tumour harbour lower mitochondrial activity and adhesion.

Localisation at the tumour edge facilitates rounded-amoeboid cell invasion into the surrounding tissue, dissemination and colonization of secondary sites. In order to disseminate, cells need to survive under low-adherence conditions in circulation. Metabolic adaptations found in rounded-amoeboid cells (Fig. 2) may provide a survival advantage in the blood stream. After survival in circulation, cancer cells need to extravasate and colonize the secondary site. Amoeboid behaviour has been previously shown to confer an advantage during lung colonization[14,15,20,32,33]. Using well-established experimental metastasis assays, we found that AMPK inhibition in A375M2 rounded-amoeboid cells reduced lung metastatic colonization (Fig. 9a and Supplementary Fig. 9c), showing that the advantage these cancer cells have in colonizing the lung relies in part on their metabolic rewiring. Finally, tumour edges at metastatic sites resemble those of primary tumours (Fig. 9b)[14]. As such, lower levels of DDR1, MFN2 and OPA1 and higher levels of pAMPK and higher amoeboid scores were also found at the IF of human melanoma metastatic lesions (Fig. 9b–d and Supplementary Fig. 9d).

Overall, our data show in vitro and in vivo that intrinsic patterns of adhesion, together with mitochondrial structure and function, govern AMPK-MYPT1-Myosin II activation to facilitate tumour dissemination.

## Discussion

Plasticity of cell migration is crucial during development, immunity and cancer dissemination. We show that adhesion levels are key for cells to sense how much traction stress to apply and how to adapt their metabolism and cytoskeleton to generate the required forces. We also show that downstream of adhesive molecules, mitochondrial dynamics control Myosin II activity via AMPK sensing (Fig. 10).

Matrix architecture influences cellular energetics[34–37] and ECM stiffness has been linked to glycolysis[36] and mitochondrial reprogramming[38] in normal epithelial cells. Also, changes in the ECM stiffness alter mitochondrial dynamics and triggers an enhanced redox response[38,39]. We describe here how AMPK and mitochondrial dynamics operate in cancer cells in physiological 3D matrices to provide the cytoskeletal plasticity required for efficient cell migration. We show that mitochondrial ATP rather than glycolysis is a distinctive feature in 3D cancer cell migration. Mitochondria undergoing fusion form tubular and interconnected networks that are associated with maximal ATP production[31]. During elongated-mesenchymal migration, high ATP levels required for establishing strong adhesion rely on highly active and fused mitochondria. Here we show that preventing DDR1-mediated adhesion, ATP levels drop, activating the metabolic sensor AMPK that in turn inactivates Myosin Phosphatase, inducing high Myosin II rounded-amoeboid migration. Importantly, AMPK activation also induces mitochondrial fission, reinforcing the energy imbalance and further boosting rounded-amoeboid migration. AMPK has been previously associated to mitochondrial dynamics under metabolic stress conditions[29,40–42]. AMPK phosphorylation of S442-MFN2 leads to non-degradative ubiquitination, inhibiting mitochondrial fusion[40,41]. Moreover, AMPK phosphorylation of MFF at S155 and S172, primes DRP1 recruitment to the mitochondrial membrane and stimulates mitochondrial fission[29]. Therefore in both cases, AMPK activation results in mitochondrial fragmentation. We report here that

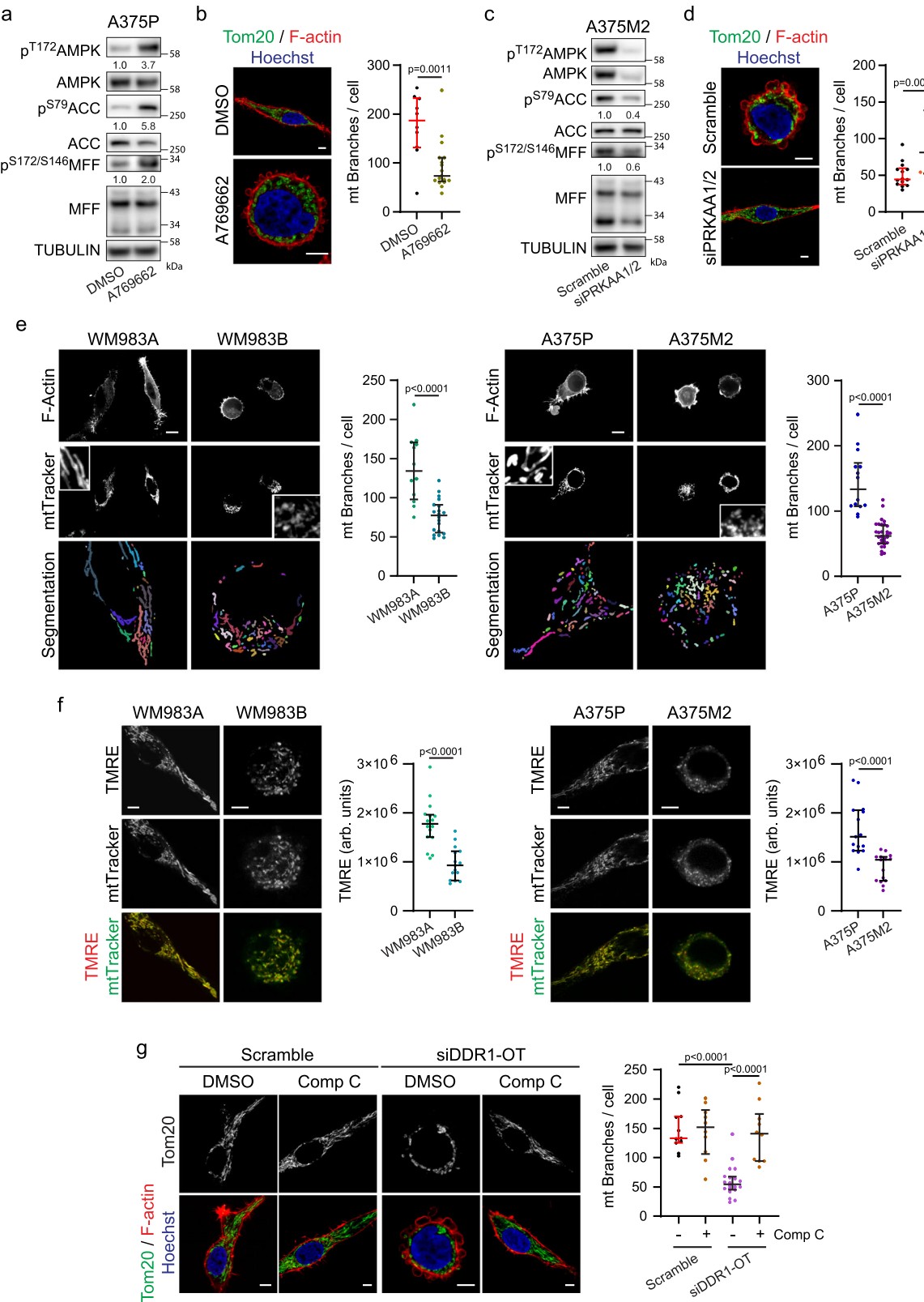

rounded-amoeboid migrating cells display higher intrinsic AMPK activation and fragmented mitochondria. In our models -without inducing any further metabolic stress- the sole impairment of mitochondrial fusion by silencing MFN2 or induction of mitochondrial fission due to AMPK-driven MFF phosphorylation led to mitochondrial fragmentation and rounded-amoeboid migration. Hypoxia has been reported to promote rounded-amoeboid migration in cancer cells[43]

and cancer cells under hypoxic conditions display fragmented and less active mitochondria[44]. Altogether, these observations suggest that rounded-amoeboid migration may be an efficient and energetically parsimonious mode of dissemination both in normoxia and under low oxygen conditions[45]. We open up the exciting possibility that mitochondria orchestrate mechano-responsiveness via AMPK-mediated ATP/AMP sensing.

**Fig. 5 | Mitochondrial dynamics in 3D migration. a** Upon AMPK activation (A769662 10 µM, 30 min) in A375P cells, western blot of the indicated proteins ($n = 3$). **b** (Left) Representative images of mitochondrial network (Tom20, green), F-actin (red) and nucleus (Hoechst, blue) after A769662 treatment (10 µM, 24 h). Scale bar = 5 µm. (Right) Quantification of mitochondrial branches per cell from Tom20 staining (10, 17 cells pooled from $n = 3$). **c** Western blot of the indicated proteins after AMPK knock-down in A375M2 cells ($n = 3$). **d** (Left) Representative images of mitochondrial network (Tom20 (green), F-actin (red) and nucleus (Hoechst, blue)) after AMPK knock-down. Scale bar = 5 µm. (Right) Quantification of mitochondrial branches per cell from Tom20 staining (14, 13 cells pooled from $n = 3$). **e** Live cell imaging of mitochondria using MitoTracker Deep Red of the indicated cell lines stably transfected with LifeAct-GFP. Bottom panel show Mito-Tracker Deep Red segmentation used for quantification. Scale bar = 10 µm. Quantification of mitochondrial branches per cell from Tom20 staining (12, 20, 14, 30 cells pooled from $n = 3$). **f** Representative images of TMRE (red) and mitoTracker Green (green). Quantification of TMRE fluorescence intensity per cell (15 cells pooled from $n = 3$). **g** (Left) Representative images of mitochondrial network (Tom20, green), F-actin (red) and nucleus (Hoechst, blue) after DDR1 knock-down and Comp C treatment (2 µM, 24 h). Scale bar = 5 µm. (Right) Quantification of mitochondrial branches per cell from Tom20 staining (11, 10, 20, 10 cells pooled from $n = 3$). Western blot quantifications normalized by each total protein (**a**, **c**). Cells seeded on a collagen I matrix (**b**, **d**, **e**, **f**, **g**). Dot plots (**b**, **d**, **e**, **f**, **g**) show median with interquartile range (each dot represents a single cell). *p* values were calculated using two-tailed tests (**b**, **d**–**f**). *p* value by unpaired *t*-test (**b**, **d**, **e**, **f**) and Kruskal–Wallis with Dunn's multiple comparisons test (**g**). All *n* are indicative of independent experiments unless otherwise stated. Source data are provided as a Source Data file. See also Supplementary Fig. 6.

Changes in mitochondrial structure often drive alterations in mitochondrial metabolism that facilitate metabolic heterogeneity[46]. A systematic pan-cancer analysis of metabolic gene expression levels suggested low mitochondrial metabolism in patients with metastatic melanoma, among other cancers[47] and mitochondrial dysfunction has been associated with increased cell migration[48]. We show here that cells with fragmented but functional mitochondria engage in lower mitochondrial metabolism, leading to AMPK activation and rounded-amoeboid invasion. Importantly, we observe that the adhesion receptor DDR1 and mitochondrial fusion proteins are down-regulated in invasive areas of human melanoma tissues while there is an increase in AMPK and Myosin II activity. Our data demonstrate that at the tumour edge, cancer cells have metabolic flexibility to invade and disseminate at a much lower energetic cost, thus posing a great danger. Moreover, this strategy would allow them to escape primary tumours where nutrients have become scarce.

Myosin II activity is regulated by a combination of direct phosphorylation of MLC2 and inactivation of Myosin Phosphatase[9,10]. Several kinases can phosphorylate MYPT, the regulatory subunit, to decrease Myosin Phosphatase activity, resulting in increased Myosin II phosphorylation/activation[10]. MYPT subunits were identified as new substrates of AMPK, the master metabolic sensor[27]. pSer472-MYPT1 promotes its interaction with 14-3-3, which impairs MYPT1 binding to Myosin II and inhibits Myosin Phosphatase activity[28]. In the present study, we show how AMPK activation in low adherent cells results in pSer472-MYPT1 and leads to increased Myosin II-dependent 3D invasion. Most studies have focused on the direct regulation of Myosin II activity by ROCK (via MLC2 phosphorylation at Thr18 and Ser19, and to a lesser extent via phosphorylation of MYPT1 at Thr696 and Thr853)[9–11]. Our observations are important to further understand the different levels of Myosin II regulation in migrating cells. While there is scarce evidence for other kinases regulating MLC2 function via MYPT in migrating cells, we suggest that AMPK phosphorylation of Ser472-MYPT1 may be a key controller of tumour cell migratory plasticity downstream of mechano-metabolic signals. However, AMPK may regulate Myosin II activity at multiple levels, as AMPK activates RhoA signalling via phosphorylation of RhoGEF2 in migrating Drosophila primordial germ cells[49].

β1 integrins are essential for both rounded-amoeboid and elongated-mesenchymal modes of movement. Despite the low degree of attachment, rounded-amoeboid melanoma cells appear to have higher levels of β1 integrins than elongated-mesenchymal melanoma cells. β1 integrins have been linked previously to rounded-amoeboid migration[1,2,50,51]. Interestingly, we show that DDR1 controls the balance between individual cell migration modes and their metabolic rewiring. Previous reports showed that DDRs enhance integrin-mediated adhesion to collagen[23,52], while loss of DDR1 increases contractility in breast tumours[53]. We propose that low DDR1 coupled to moderate levels of β1 integrins provide the required degree of adhesion to collagen for cancer amoeboid 3D migration.

We have previously shown that Myosin II activity provides pro-survival signals in cancer cells under drug treatments[15] or under anoikis-inducing conditions[33]. To metastasize, cells leave the primary tumour and need to survive in circulation. Matrix detachment during dissemination has been linked to metabolic rewiring[54] and Myosin II activity protects circulating cancer cells from shear stress in the blood[55]. We show here how cells in suspension, similar to cells in circulation, exhibit lower OXPHOS levels. Furthermore, we describe how rounded-amoeboid migration is also characterised by low levels of OXPHOS and their metastatic colonization abilities rely on AMPK signalling. We speculate that by lowering their mitochondrial metabolism, circulating tumour cells increase their Myosin II activity, therefore promoting cancer cell survival in low-adherence environments. We propose that rounded-amoeboid cells are primed for dissemination and metastasis by entering a low-energy requirement mode that may allow them to survive under stressful conditions. Understanding the mechanisms underlying cell migration plasticity is essential to unravel new vulnerabilities of cancer metastasis with therapeutic potential. AMPK dependence might therefore be an Achilles heel in highly-metastatic rounded-amoeboid cells. However, AMPK plays different roles in tumorigenesis, enabling metabolic adaptation under specific stress conditions such as hypoxia or glucose deprivation, favouring cancer cells survival but also suppressing cancer cell proliferation and tumour formation mainly via mTOR regulation[56]. Further studies are required to clarify when and how AMPK antagonists will be beneficial to prevent cancer dissemination. In addition, our data suggest that the usage of compounds that directly or indirectly activate AMPK in other pathologies such as diabetes, mitochondrial disease and cardiovascular diseases[57], might need to be carefully evaluated, particularly in those patients with advanced cancer.

Overall, our work unveils how mitochondrial dynamics control the cytoskeleton and cell migration and place AMPK as a key mechano-metabolic sensor during tumour dissemination.

## Methods

### Experimental model and subject details

This research complies with all relevant ethical regulations.

**Animal studies.** All animals were maintained under specific pathogen-free conditions and handled in accordance with the Institutional Committees on Animal Welfare of the UK Home Office (The Home Office Animals Scientific Procedures Act, 1986). Animals were housed in the QMUL Biological Services holding facility, which maintained a 7 h light/dark cycle, an ambient temperature of 19–22 °C and humidity of 50–60%. All animal experiments were approved by the Ethical Review Process Committees at Barts Cancer Institute and King's College London, in accordance with the Animals (Scientific Procedures) Act 1986 and according to the guidelines of the Committee of the National Cancer Research Institute. Gender

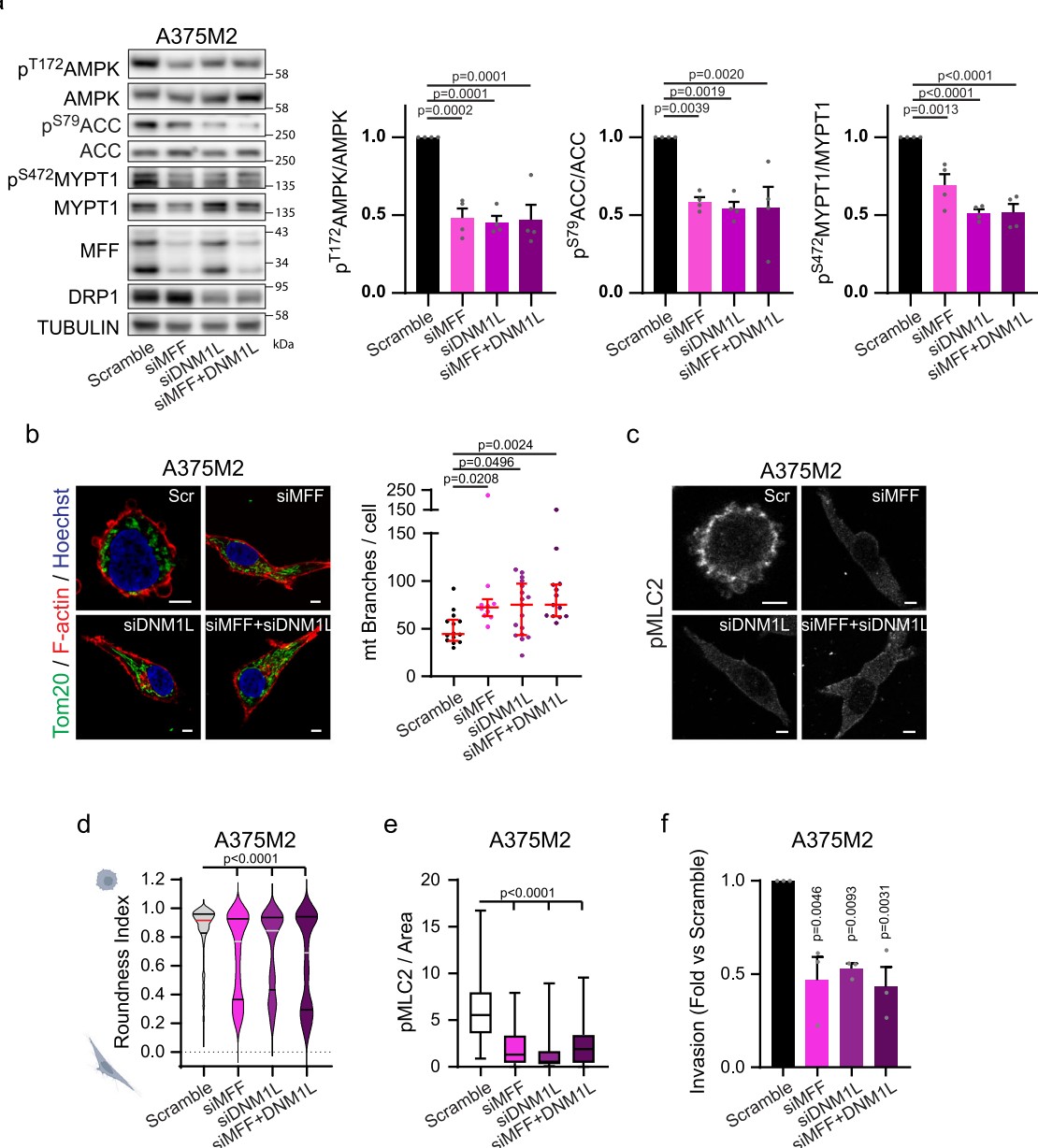

**Fig. 6 | Mitochondrial fission control the cytoskeleton and invasion. a** Western blot (left) and quantification (right) of the indicated proteins after the indicated knock-downs in A375M2 cells ($n = 4$). **b–e** Cells seeded on a collagen I matrix after the indicated knock-downs. **b** (Left) Representative images of mitochondrial network (Tom20 (green), F-actin (red) and nucleus (Hoechst, blue)). Scale bar = 5 μm. (Right) Quantification of mitochondrial branches per cell from Tom20 staining (14, 10, 17, 13 cells pooled from $n = 3$). **c** pMLC2 staining for the indicated conditions, scale bar = 5 μm ($n = 3$). **d–f** After the indicated knock-downs, quantification of cell morphology (211, 231, 222, 169 cells pooled from $n = 3$) (**d**), quantification of pMLC2 immunofluorescence signal normalized by cell area (52, 53, 64, 71 cells pooled from $n = 3$) (**e**) and 3D invasion index into a collagen I matrix ($n = 3$) (**f**). Dot plot (**b**) shows median with interquartile range (each dot represents a single cell). Violin plot (**d**) shows median with interquartile range. Box plot (**e**) shows median (centre line), interquartile range (box) and min-max values (whiskers). Graphs (**a, f**) show mean ± SEM. $p$ value by Kruskal–Wallis with Dunn's multiple comparisons test (**b, d, e**) and one-way ANOVA with Dunnett's correction (**a, f**). All $n$ are indicative of independent experiments unless otherwise stated. Source data are provided as a Source Data file. See also Supplementary Fig. 7.

was in line with previous work from our lab using melanoma cell lines.

**Animal derived tissues.** Tumours from WM983A-, WM983B-, A375P- and A375M2-EGFP cells injected subcutaneously into severe combined immunodeficient mice (SCID; CB17/Icr-Prkdcscid/IcrIcoCrl) were used for immunohistological purposes. Tumours were generated in a previous study[14].

**Lung colonisation assay.** For experimental metastasis assays, A375M2 cells pre-treated in vitro for 24 h with either DMSO or Compound C

(2 μM) were labelled with 10 μM CMFDA-Green (C7025, Life Technologies) for 10 min and then trypsinized and counted. $1 \times 10^6$ labelled cells / 0.2 ml PBS along with drugs (same concentration as pre-treatment) were injected into tail vein of 9–10 week old female NOD/SCID/IL2Rγ-/- mice (NSG, Charles River). Mice were sacrificed 30 min (to confirm that equal numbers arrived at the lung) and 24 h after tail vein injection. Lungs were extracted, washed with PBS (with calcium/magnesium) twice and fixed with 4% formaldehyde for 16 h at 4 °C. Lungs were examined under a Zeiss LSM 710 Meta confocal microscope (Carl Zeiss) with a 20X objective. Data are presented as percentage of field of area covered by fluorescence, and 15 fields per mouse were analysed.

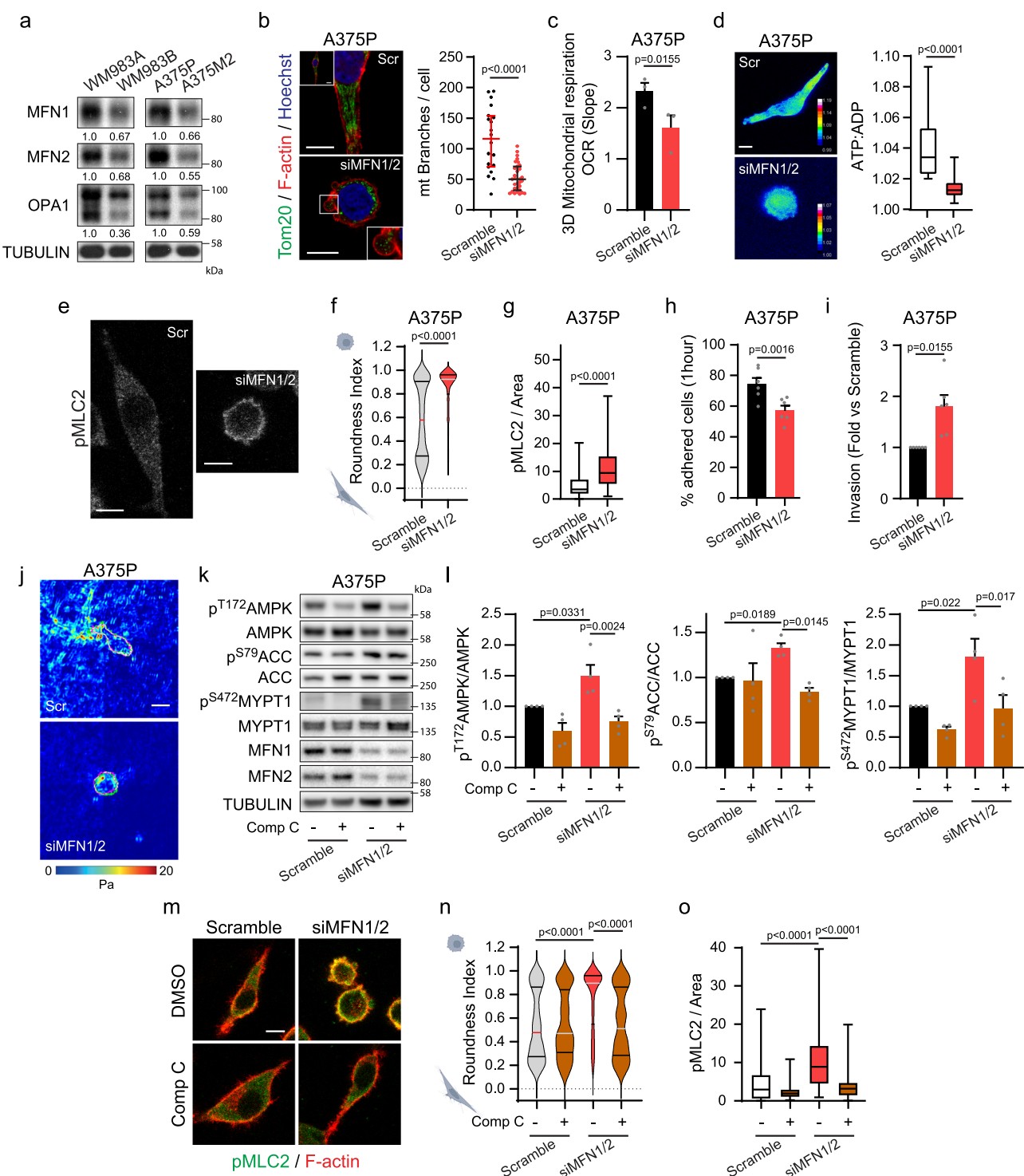

$n = 7$ mice/condition for each experiment (2 mice sacrificed 30 min and 5 mice 24 h after tail vein injection), $n = 2$ independent experiments.

**Patient derived tissues.** Tumours were classified following the most recent World Health Organization criteria. Tumour samples were processed by IRBLleida (PT17/0015/0027) and HUB-ICO-IDIBELL (PT17/0015/0024) Biobanks integrated in the Spanish National Biobank Network and Xarxa de Bancs de Tumors de Catalunya following standard operating procedures with the appropriate approval of the Ethics and Scientific Committee. Samples were collected with specific informed consent, in accordance with the Helsinki Declaration. Two tissue microarrays including FFPE biopsies of 46 human primary melanomas and 45 metastasis were included in the case series (for clinical information see Supplementary Tables 1 and 2). Each biopsy was represented by two cores (1 mm diameter) from the tumour body (TB) and two cores from the invasive front (IF) areas.

**Cell lines.** HT1080 (CVCL_0317) fibrosarcoma cells were from Prof Chris Marshall (ICR, UK), MDA-MB-231 breast cancer cells were from Prof Clare Isacke (ICR, UK), A375P (CVCL_6233) and A375M2 (CVCL_C0RP) cells were from Prof Richard Hynes (HHMI, MIT, USA) and WM983A (CVCL_6808) and WM983B (CVCL_6809) were purchased from Coriell Institute (USA). HEK293T (CVCL_0063) cells were from Dr. Jeremy Carlton (The Francis Crick Institute, UK). All cell lines

**Fig. 7 | Mitochondrial fusion control the cytoskeleton and invasion.**
**a** Mitochondrial fusion protein levels ($n = 4$). Quantifications normalized versus TUBULIN. **b** (Left) Mitochondrial network (Tom20 (green), F-actin (red) and nucleus (Hoechst, blue)) after MFN1/2 knock-down. Inset shows whole cell (Scramble) and bleb protrusion (siMFN1/2). Scale bar = 10 μm. (Right) Quantification of mitochondrial branches per cell from Tom20 staining (20, 31 cells pooled from $n = 3$). **c** Oxygen consumption rate (OCR) from cells in 3D collagen ($n = 3$). **d** Maximal intensity projection (left) and quantification (right) of ATP:ADP ratio in cells expressing PercevalHR biosensor, in a 3D collagen matrix (27, 32 cells pooled from $n = 3$). Scale bar = 5 μm. **e–j** After MFN1/2 knock-down, pMLC2 immunofluorescence ($n = 3$), scale bar = 10 μm (**e**); quantification of cell morphology (383, 356 cells pooled from $n = 3$) (**f**), pMLC2 immunofluorescence signal normalized by cell area (116, 130 cells pooled from $n = 3$) (**g**), adhered cells after 1 h seeding on a collagen I matrix ($n = 6$) (**h**), 3D invasion index into a collagen I matrix ($n = 6$) (**i**) and representative traction stresses exerted by cells ($n = 3$) (**j**). Colour bar indicates traction stress magnitude (Pascal, Pa), scale bar = 20 μm. **k, l** Western blot and quantification after MFN1/2 knock-down and Comp C treatment (2 μM, 24 h) ($n = 4$). **m** Cells seeded on a collagen I matrix. pMLC2 (green) and F-actin (red) after MFN1/2 knock-down and Comp C treatment (2 μM, 24 h) ($n = 3$). Scale bar = 10 μm. **n** Cell morphology (299, 222, 276, 238 cells pooled from $n = 3$). **o** pMLC2 immunofluorescence signal normalized by cell area (82, 75, 86, 79 cells pooled from $n = 3$). Dot plot (**b**) shows median with interquartile range (each dot represents a single cell). Violin plots (**f, n**) show median with interquartile range. Box plots (**d, g, o**) show median (centre line), interquartile range (box) and min-max values (whiskers). Graphs (**c, h, i, l**) show mean ± SEM. $p$ values calculated using two-tailed tests (**b–d, f–i**). $p$ value by one sample $t$-test unpaired $t$-test, Mann-Whitney test (**d, f, g**), Paired $t$-test (**c, h**), one-way ANOVA with Dunnett's correction (**l**) and Kruskal-Wallis test with Dunn's multiple comparisons test (**n, o**). All $n$ are indicative of independent experiments unless otherwise stated. Source data are provided as a Source Data file. See also Supplementary Fig. 7–8.

were grown at 37°C and 10% $CO_2$ in DMEM Supplemented with 10% FBS and 1% penicillin/streptomycin (all from GIBCO). A375P, A375M2, WM983A and WM983B were authenticated using short tandem repeat DNA profiling. All cell lines were routinely tested for mycoplasma contamination. All cell lines were kept in culture for a maximum of three to four passages and cell phenotypes were verified routinely.

For actin imaging in live cells, cells stably transfected with a LifeAct-GFP plasmid (provided by Dr Rikki Eggert, KCL, UK) were used.

## Chemicals
Chemicals used in this study: Myosin II inhibitor blebbistatin (Cat. 203390, Calbiochem; resuspended in 95% DMSO; used at 25 μM), antimycin A (Cat. 15405739, Fisher Scientific; resuspended in DMSO; used at 0.5 μM and 1 μM), oligomycin (Cat. O4876, Sigma-Aldrich; resuspended in DMSO; used at 1 μM), rotenone (Cat. R8875, Merck; resuspended in DMSO; used at 0.5 μM), FCCP (Cat. 0453, Tocris Bioscience; resuspended in DMSO; used at 0.75 μM), Compound C (Cat. 171260, Sigma-Aldrich; resuspended in DMSO; used at 2 μM) and A769662 (Cat. 3336, Tocris Bioscience; resuspended in DMSO; used at 10 μM).

## Antibodies
Antibodies and concentrations used: pSer19-MLC2 (#3671; 1:200 immunofluorescence; 1:50 immunohistochemistry), DDR1 (#5583; 1:1000 immunoblot, 1:200 immunohistochemistry), DRP1 (#8570; 1:1000 immunoblot), MFF (#86668; 1:1000 immunoblot), MFN1 (#14739; 1:1000 immunoblot), pThr172-AMPK (#2535; 1:1000 immunoblot; 1:150 immunohistochemistry), AMPK (#2532; 1:1000 immunoblot), MYPT (#8574; 1:1000 immunoblot), pSer79-Acetyl-CoA Carboxylase (#3661; 1:1000 immunoblot), Acetyl-CoA Carboxylase (#3662; 1:1000 immunoblot), from Cell Signaling Technology; pSer172/Ser146-MFF (AF2365, 1:1000 immunoblot (Ser172 in MFF human isoform 1 corresponds to Ser146 in MFF human isoforms 2, 4 and 5)), pSer472-MYPT1 (AF3779; 1:1000 immunoblot), from Affinity Biosciences; GAPDH (MAB374; 1:10000 immunoblot) from Merck-Millipore; Tubulin (T6199; 1:10000 immunoblot) from Sigma-Aldrich; Integrin β1 (ab24693; 1:100 immunofluorescence), MFN2 (ab56889; 1:1000 immunoblot, 1:500 immunohistochemistry), CD44 (ab157107; 1:700 immunohistochemistry) from Abcam; OPA1 (612606; 1:1000 immunoblot, 1:500 immunohistochemistry) from BD Transduction Laboratories; Tom20 (sc-17764; 1:800 immunofluorescence) from Santa Cruz Biotechnology.

## Cell culture on 3D collagen I matrices
Fibrillar bovine dermal collagen (PureCol, Cat. 5005 Advanced Bio-Matrix) was prepared at 1.7 mg/ml in DMEM (300 μl/well for a 24-well plate; 100 μl/well for a 96-well plate). After collagen polymerization (4 h at 37°C – 10% $CO_2$), cells were seeded on top in DMEM 10% FBS, allowed to adhere for 24 h and treatments added (where appropriate).

To embed cells within the collagen matrix, cells were suspended in the collagen and the appropriate volume of the suspension was seeded (100 μl/well for a 96-well plate; 250 μl/well for optical bottom 8-well μ-slide). The collagen matrices containing cells were allowed to polymerize for 4 h at 37°C – 10% $CO_2$. Then DMEM 10% FBS was added on top. All assays were performed with cells seeded on top of a thick layer of collagen unless otherwise mentioned.

## siRNA transfection
Reverse transfection was used to transiently down-regulate the indicated genes. Two hundred and fifty thousand cells were seeded in complete media (DMEM 10%FBS) in 6-well plates with a mix containing 20 nM siGENOME SMARTpool or individual On-Target (OT) siRNA oligonucleotides (Dharmacon), Optimem-I and Lipofectamine 2000 (Invitrogen). Non-targeting siRNA were used as control. 24–48 h after transfection cells were split and seeded for the downstream experiments/treatments. All siRNA sequences were from Dharmacon (Lafayette, USA) and are listed in Supplementary Table 3.

## Immunoblotting
Cells were lysed in Laemmli Lysis Buffer and snap frozen. Then, lysates were boiled for 5 min, sonicated for 15 s and spun down. Cell lysates were resolved by SDS-polyacrylamide gels (SDS-PAGE) in non-reducing conditions and transferred to PVDF filters (0.45 μm, Immobilion™). Membranes were blocked in 5% BSA in 0.1% Tween 20-TBS. Primary antibodies were incubated overnight at 4°C. ECL Plus or Prime ECL detection System (GE Healthcare) with HRP-conjugated secondary antibodies (Amersham ECL Rabbit IgG, HRP-linked whole Ab (from donkey) NA934 and Amersham ECL Mouse IgG, HRP-linked whole Ab (from sheep) NA931, 1:10 000 GE Healthcare) were used for detection. Bands were quantified using Fiji 1.53t software (http://fiji.sc).

## Collagen imaging
Imaging was performed with a Zeiss LSM 510 Meta confocal microscope (Carl Zeiss) with C-Apochromat X 40/1.2 NA (water) objective lenses and Zen software (Carl Zeiss). Reflectance imaging of the matrix was performed by collecting the backscattered light. Since reflectance is a surface property, extreme caution was taken to ensure the acquisition of images at the same height across all conditions.

**Points of attachment.** Collagen I was imaged via reflectance microscopy and points of attachment were assessed by counting the points of collagen attachment to the actin cytoskeleton (visual assessment) through consecutive z-slices. The points of attachment were manually counted using the Cell counter tool.

## Measurement of traction stress
Live, 3D time-lapse videos coupled with image-processing were used to measure stress applied by cells within 3D collagen environments.

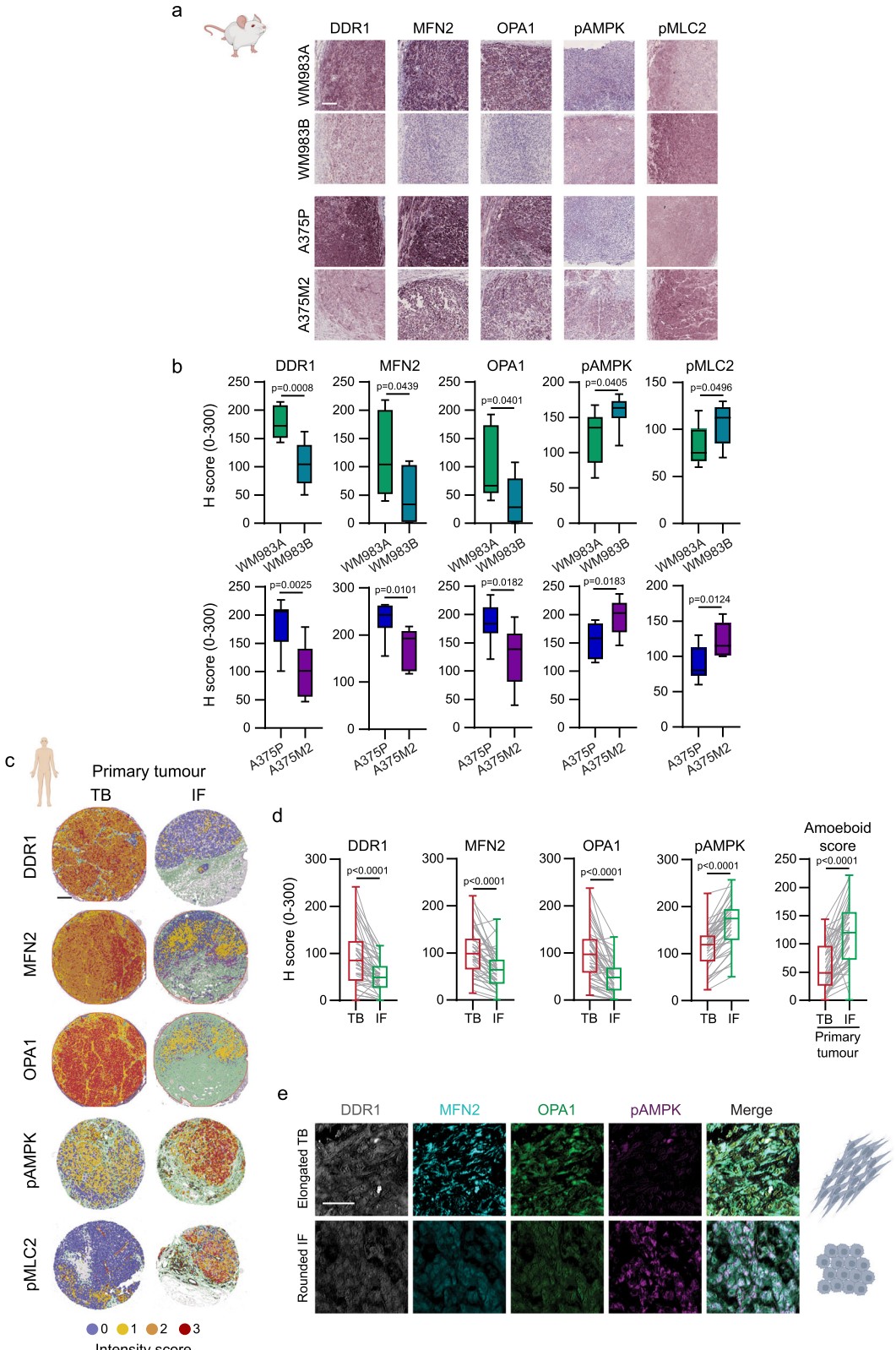

**Fig. 8 | Adhesion, AMPK and mitochondrial dynamics in vivo and in primary melanoma tissues. a**, **b** Representative immunohistochemistry (IHC) images and quantification of DDR1, MFN2, OPA1, pAMPK and pMLC2 in subcutaneous tumours in mice from WM983A, WM983B, A375P or A375M2 cells (*n* = 8 mice/group). Scale bar = 100 μm. **c**, **d** Representative QuPath mark-up images and quantification of DDR1, MFN2, OPA1 and pAMPK expression and amoeboid score (calculated as indicated in methods) in matched tumour body (TB) and invasive front (IF) of primary tumours from human melanoma tissue microarrays (*n* = 46 tumours). Scale bar = 200 μm. **e** Representative IHC images showing DDR1, MFN2, OPA1 and pAMPK staining in elongated cells from TB and rounded cells from IF. Scale bar = 50 μm. Box plots (**b**, **d**) show median (centre line), interquartile range (box) and min-max values (whiskers). *p* values were calculated using two-tailed tests (**b**, **d**). *p* value by unpaired *t*-test (**b**) and Wilcoxon test (**d**). Source data are provided as a Source Data file. See also Supplementary Fig. 9.

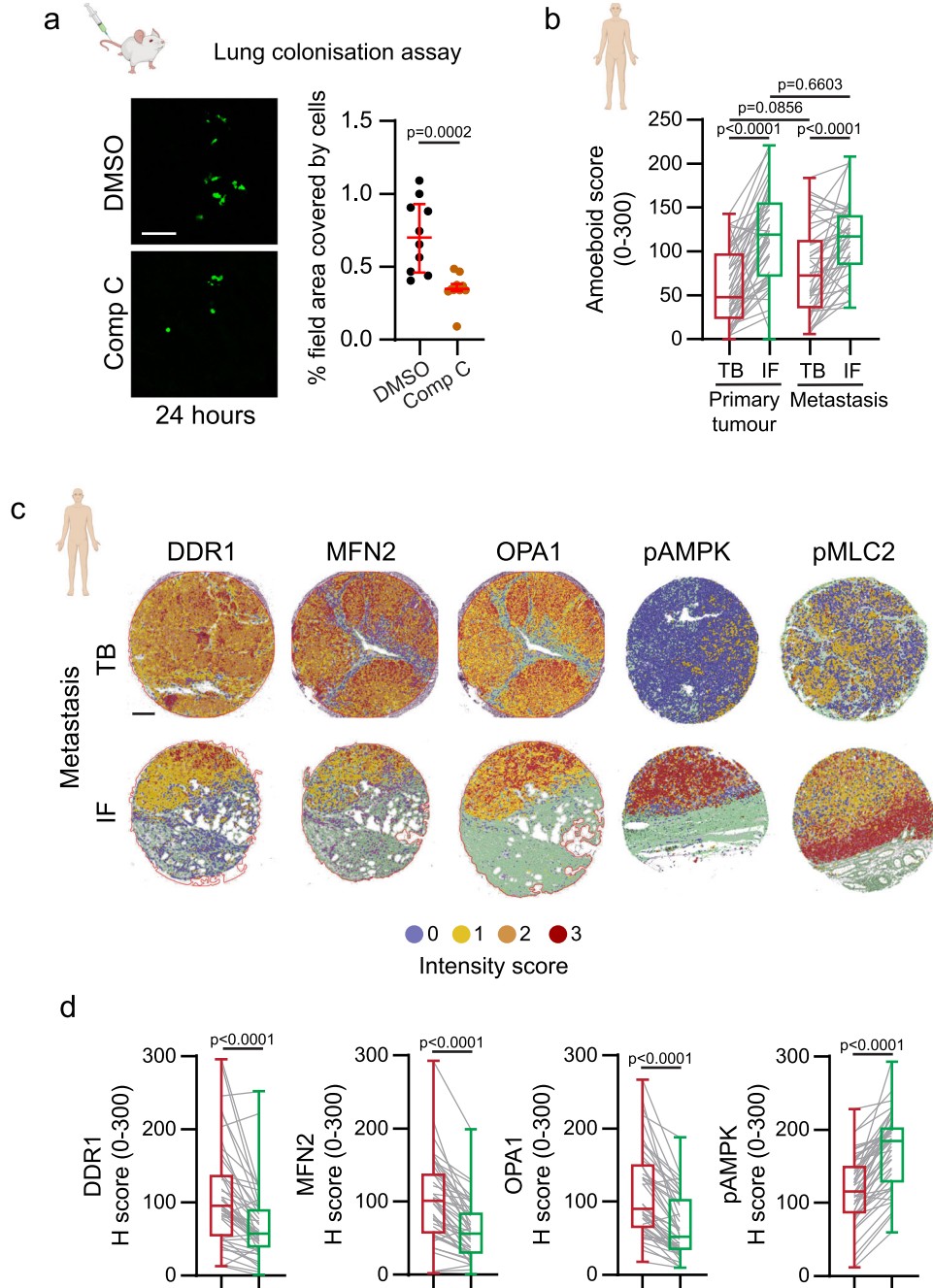

**Fig. 9 | Adhesion, AMPK and mitochondrial dynamics in metastasis. a** (Left) Confocal images of mouse lungs 24 h after tail vein injection of 5-chloromethylfluorescein diacetate (CMFDA)-Green labelled A375M2 pre-treated with Comp C (2 µM, 24 h) and (right) percentage of field area covered by cells (15 fields/mouse/condition, 5 mice/condition, *n* = 2 independent experiments). Scale bar = 100 µm. **b** Amoeboid score comparing tumour body (TB) and invasive front (IF) in melanoma primary tumours and metastasis, calculated as indicated in methods (primary tumours *n* = 46, metastasis *n* = 42). **c, d** Representative QuPath mark-up images and quantification of DDR1, MFN2, OPA1, pAMPK and pMLC2

expression in matched TB and IF of metastasis from human melanoma tissue microarrays (*n* = 45 patients). Scale bar = 200 µm. Dot plot (**a**) shows median with interquartile range (each dot represents a single mouse). Box plots (**b, d**) show median (centre line), interquartile range (box) and min-max values (whiskers). *p* values were calculated using two-tailed tests (**a, d**). *p* value by Mann-Whitney test (**a**), one-way ANOVA with Holm-Šídák's multiple comparisons test (**b**) and Wilcoxon test (**d**). Source data are provided as a Source Data file. See also Supplementary Fig. 9.

Cells labelled with LifeAct-GFP were embedded within 3D collagen I matrices polymerized in tissue culture wells with optical plastic bottom and imaged directly in-well using an inverted confocal microscope, for 8 h. Cells were seeded such that the cellular density was sparse enough such that only one cell was present within an area of at least 250 × 250 x 50 µm in the middle of the 3D collagen matrix. This was done to ensure that the displacements measured within a certain field were only due to stress generated by the cell within the field of view.

Collagen I was imaged by collecting the reflected/backscatter light from the matrix using laser-scanning confocal microscopy[19,20,58]. This approach was selected due to ease of access and lack of alterations/artificial manipulations to the matrix. By comparison of collagen I immunostained with an anti-collagen I antibody, it has been

## Metabolic and Cytoskeletal Adaptation

**Fig. 10 | AMPK is a mechano-metabolic sensor linking cell adhesion and mitochondrial dynamics to Myosin II dependent cell migration.** Highly invasive but weakly adhesive rounded-amoeboid cancer cells harbour high levels of cortical Myosin II activity and exert low magnitude traction forces into extracellular matrix (ECM). Low adhesion to the matrix results in decreased engagement in mitochondrial metabolism and alterations in ATP and AMP intracellular levels that lead to AMP-activated protein kinase (AMPK) activation. Once AMPK is active, it directly phosphorylates the crucial regulator of cytoskeletal dynamics Myosin phosphatase (Myosin phosphatase target subunit 1, MYPT1) in the key residue for its inactivation, leading to increased Myosin Light Chain phosphorylation and increased overall Myosin II activity. At the same time, AMPK induces mitochondrial fission through the phosphorylation of Mitochondrial Fission Factor (MFF), which sustains the imbalance in energy levels, further boosting AMPK signalling and Myosin II activation. In contrast, highly adhesive elongated-mesenchymal cells present highly fused and active mitochondria. This provides energy for mesenchymal cancer cells to exert strong adhesive traction stress, while it maintains lower levels of AMPK signalling, resulting in moderate Myosin II activity. Strong adhesions in elongated-mesenchymal cells rely on Discoidin Domain Receptor 1 (DDR1) collagen receptor. Reducing DDR1-dependent adhesion, inhibiting mitochondrial fusion or inducing AMPK activity in elongated-mesenchymal cells promotes the transition to rounded-amoeboid efficient migration/invasion and all its cytoskeletal/mitochondrial features. (Created with BioRender.com).

demonstrated that there is negligible detection error from reflectance imaging of collagen I[19]. The detection error arises from backscatter-negative fibrils in vertical orientation and is below 3%.

**Computation of displacements and strains.** Videos were generated by acquiring 3D image stacks of 30 μm thickness with step size of 1 μm. The optimal interval for acquisition of images was determined to be intervals of 2 min to avoid phototoxicity. The resultant videos allowed for resolution of individual collagen I fibres surrounding each cell within the x-y plane, which represents the migration of LifeAct-GFP expressing cell (mid-plane) within the surrounding matrix over 10 min.

This set-up allowed for the optimal resolution of collagen fibres within the z-plane. The videos were then processed using a custom-written Particle Image Velocimetry (PIV) based tracking algorithm in Wolfram Mathematica 10. In order to track the matrix, 3D videos were deconstructed and analysed on a frame-by-frame basis. The deconstruction of the videos resulted in a 512 x 512 x 90 image stack corresponding to each time point. The 90 positions correspond to the interpolation along the z-axis. Superimposition of a mesh grid on the image allowed for the tracking of the position of each pixel in one frame, with respect to its relative position in the adjacent frame, at the next time point in 2D x-y plane and, in the 3D case, through each slice of the z-stack. The

difference between these positions was used to calculate displacement for all three dimensions. In cases where the corresponding point in an adjacent frame could not be found, the region would be labelled zero and would then be linearly interpolated to the nearest 9 neighbouring pixels. After analysis in all three planes $(xy, xz, yz)$ the 3D axes were then rotated and adjusted in order to reconstruct the 3D displacement data. Cell displacements calculated in $X$, $Y$ and $Z$ corresponding to each pixel in the 512 x 512 x 90 frame stacks, were converted to X, $Y$ and $Z$ strain values using the inbuilt derivative filter function DerivativeFilter, in Mathematica. The conversion of displacements to strains was based on the displacement-gradient technique[59].

**Cell traction stress computation.** In order to calculate stress from strains, the stiffness of the bovine collagen matrix was calculated using AFM microscopy. Stiffness of bovine collagen was ~19 Pa, comparable to the stiffness observed in other studies[19,60]. We assume the material is purely elastic, continuous, homogeneous and isotropic. Stresses were calculated by integrating the strain data, depth information and stiffness of the matrix, using a direct Traction Force Microscopy (TFM) method[61], as opposed to more complicated and computationally extensive methods involving the solution of inverse equations and boundary conditions. The mathematical approach was based on that used by[18], following the assumption of a linearly isotropic material or substrate. According to this assumption, the Cauchy relation for traction stress, τ, is:

$$\boldsymbol{\tau} = \boldsymbol{\delta}.\boldsymbol{n} \qquad (1)$$

where δ is the Cauchy stress tensor and n the direction of the normal vector.

The code was designed to process each of the aforementioned 512 variables, setting the $X$, $Y$ and $Z$ strain values as the principal strains (diagonal elements) of a 3D symmetric strain tensor $\boldsymbol{\varepsilon}$, defined as:

$$\boldsymbol{\varepsilon} = \begin{bmatrix} \varepsilon_{11} & \varepsilon_{12} & \varepsilon_{13} \\ \varepsilon_{21} & \varepsilon_{22} & \varepsilon_{23} \\ \varepsilon_{31} & \varepsilon_{32} & \varepsilon_{33} \end{bmatrix} \qquad (2)$$

Eq. (3) below was solved in order to obtain the shear modulus $\mu$, which was then input into Eq. (4), to obtain the stress tensor $\boldsymbol{\delta}$, for which isotropic linearly elastic properties are assumed[60]. The constitutive mechanical properties required for these calculations included a Young's modulus $E$ of 28 Pa, measured for our collagen matrix, and a Poisson ratio $v$ of 0.25 based on that measured by Steinwachs et al. (2016).

$$E = 2\mu(1 + v) \qquad (3)$$

$$\boldsymbol{\delta} = 2\mu\boldsymbol{\varepsilon} \qquad (4)$$

The eigen Eq. (5) was then solved for the three eigenvalues *(λ1, λ2, λ3)* and eigenvectors $(\mathbf{V_1}, \mathbf{V_2}, \mathbf{V_3})$ of this stress tensor. The eigenvalues correspond to the traction stress solutions of the stress tensor $\boldsymbol{\delta}$, whilst the eigenvectors correspond to the direction of the normal $\mathbf{n}$, relative to the surface on which the traction stress is acting.

$$\mathbf{E} = (\boldsymbol{\delta} - \lambda\mathbf{I})\mathbf{V} \qquad (5)$$

where E represents the set of vectors which satisfy $(\boldsymbol{\delta} - \lambda 1)$ **V**=0.

The matrix dot product of the stress eigen values with their correct and corresponding eigen vectors, yields the traction stress $\boldsymbol{\tau}$, in the Cauchy relation (Eq. 1).

Once the traction stresses had been computed in $X$, $Y$ and $Z$ $(T_x, T_y, T_z)$, the magnitude of these stresses was computed using the standard vector combination relation:

$$|\mathbf{T}| = \sqrt{T_x^2 + T_y^2 + T_z^2} \qquad (6)$$

After processing each data stack, a 512 × 512 pixel colour map was generated for each slice in the stack using a jet LUT. According to this colour scheme, the resulting gradient based stress maps were colour coded such that the red areas of the reaction maps correspond to traction stress while blue regions correspond to minimal or zero traction stress and are found in areas of matrix where no substantial impact due to cell movement was detected.

## Cell morphology on collagen
Still phase-contrast images of cells seeded on collagen matrices were acquired with Zeiss Axio Vert. A inverted microscope and morphology was quantified on using ImageJ. Cell morphology was assessed using the morphology descriptor tool roundness after manually drawing around the cell. Values closer to 1 represent rounded morphology; values closer to 0 represent more spindle-shaped cells.

## Cell adhesion to collagen I
Collagen matrices were prepared in 96-well plates. Ten thousand cells/well were seeded in duplicate or triplicate in the 96-well plates and incubated at 37 °C for the indicated times. The centre of the well was then imaged with a phase contrast microscope, washed three times with PBS and imaged again with the same settings. Each well was imaged before and after washing. Cells were counted manually using Cell counter function in ImageJ and results are presented as percentage of adhered cells, calculated as number of cells after washing divided by the number of cells before washing.

## 3D invasion
For 3D invasion assays, cells were resuspended in serum-free bovine collagen I solution at 2.3 mg/ml to a final concentration of 14,000 cells per 100 µl of matrix and spun down, in a 96-well plate. After the matrix was polymerized, 10% FBS-containing media was added on top of the matrix. After 24 h cells were fixed, stained with Hoechst and imaged using a Zeiss LSM 710 confocal microscope. Invasion was calculated as number of invading cells at 50 µm divided by the number of cells at the bottom.

## Immunofluorescence and confocal imaging
Cells seeded on top of collagen matrices were fixed with 4% formaldehyde for 15 min at room temperature. Then, cells were permeabilised for 20 min using 0.3% Triton X-100 in 5% BSA-PBS, blocked in 5% BSA-PBS for 30 min and immunostained with primary antibody overnight at 4 °C. Next, samples were incubated with Alexa Fluor™ 546-phalloidin (A22283, Thermo Fisher, 1:400) and Goat anti-rabbit Alexa Fluor™ 488 (A-11008, Thermo Fisher, 1:1000) or Goat anti-mouse Alexa Fluor™ 488 (A-11029, Thermo Fisher, 1:1000) secondary antibodies for 2 h at room temperature. Nuclei were stained with Hoechst prepared in PBS. Antibodies were diluted in 5% BSA-PBS.

Images were taken with a Zeiss LSM 510 Meta confocal microscope with Plan-Apochromat 40x/1.2 NA (water) objective lenses, Zeiss LSM 710 confocal microscope with Plan-Apochromat 40x/1.3 Oil DIC M27 and Zeiss LSM 880 confocal microscope with Airyscan super-resolution mode and Plan-Apochromat 63x/1.4 Oil DIC M27 objective lenses (Carl Zeiss, Germany). Zen software was used to acquire images (Carl Zeiss, Germany). Images were analysed using ImageJ software (NIH). For pMLC2 quantification, fluorescence signal was quantified by calculating the area occupied by pMLC staining in single cells relative to the cell area. For pMLC2 distribution, line scan analysis was performed in Fiji 1.53t using the Plot profile plug in.

## Mitochondrial mass and mitochondrial activity

Mitochondrial activity and mitochondrial mass comparing adherent versus floating cells were analysed by flow cytometry. Cells were seeded either on regular cell culture plates or polyhema pre-coated plates (for non-adherent conditions). After 24 h cells were collected, spun down and resuspended with a solution containing MitoTracker Deep Red (M22426, Thermo Fisher) (25 nM) and TMRE (ab113852, Abcam) (200 nM), at 37 °C for 20 min. After, FACS buffer (1% BSA, 2 mM EDTA and 0.1% NaN3 in PBS) was added to samples and immediately analysed on a BD FACS CANTO II flow cytometer. Data were analysed using FlowJo 10.6 software (Tree Star). A representation of the gating strategy used is shown in Supplementary Fig. 2j.

Mitochondrial activity and mitochondrial mass comparing cells on top of collagen were analysed by confocal imaging. Live cells expressing LifeAct-GFP seeded on top of a collagen I matrix were incubated with MitoTracker Deep Red (25 nM) for mitochondria visualisation and with tetramethylrhodamine ethyl ester (TMRE; 200 nM) for mitochondrial activity analysis, at 37 °C for 20 min. Then, were kept in FluoBrite DMEM for imaging. Z-stack images were taken with a Nikon Eclipse Ti Inverted equipped with a Yokogawa CSU-X1 Spinning Disk unit 100x oil objective lenses. Images were analysed using ImageJ software (NIH). Data were confirmed using MitoTracker Green in non-Life Act-GFP cells, to ensure that MitoTracker signal was not affected by the differences observed in mitochondrial membrane potential. Mitochondrial activity (TMRE signal) was quantified from maximal intensity projections from a mask created using mitochondrial staining from MitoTracker signal (MitoTracker Deep Red or Mitotracker Green (M7514, Thermo Fisher)).

## Mitochondrial morphology

For mitochondrial circularity analysis, MitoTracker Deep Red z-stacks images were imported in to Mathematica 11, converted to 3D and decomposed in to frequency bands using difference of Gaussian filters using radii ranging from 1-to-32 in powers of 2. This enabled simultaneous smoothing and background subtraction. The frequency-decomposed images were then combined to create a frequency-encoded image. The images were then Top-hat transformed with a radius of 4 to highlight the mitochondria and reduce uneven intensities. Filtered images where then thresholded and binary objects that were less than ten pixels in size were deleted. The internal Mathematica function Component Measurements was then used to calculate the circularity of each object of the image. These were then exported as csv files for further analysis.

For the analysis of mitochondrial branches per cell, a semi-automated image-based analysis was performed in ImageJ using Mitochondria Analyzer[62]. Single plane images of Tom 20 fluorescently-labelled mitochondria in cells grown on a collagen I matrix were pre-processed and converted to 8-bit images. Next, 2D Optimize Threshold command was used to identify the appropriate settings on test samples from each set acquired under similar imaging conditions. Block size of 2.25 μm and C-value of 28 was used for the 2D Analysis in batch mode. Analysis was obtained on a per-cell basis.

## NMR metabolomics

The metabolic profiles were assessed using NMR metabolomics. Three million cells were seeded in 14.5 cm dishes 48 h prior to dual-phase metabolite extraction. Media was removed and cells were washed twice with cold PBS. Then, each dish was placed on dry ice for 2 min to quench the metabolome. Afterwards, 2.2 mL of cold methanol was added and cells were scraped and transferred to a pre-cooled 15 mL tube. 2.2 mL of cold chloroform was added to each tube and mixed in a tube rotor for 10 min at 4 °C. Then, 2.2 mL of milli-Q water was added to each tube and the contents were mixed by inversion and incubated on ice for 10 min to allow the formation of a stable bilayer. Samples were then centrifuged at 4 °C and 450 g for 45 min to separate the top

layer and a bottom layer, containing metabolites soluble in the polar and apolar fraction, respectively. The polar fraction was dried in an Eppendorf concentrator (Concentrator plus, Eppendorf).

Prior to NMR experiments, samples were resuspended in a 90/10 $H_2O/D_2O$ buffer, with 100 mM $Na_2HPO_4$, 5 mM TSP and 4 mM $NaN_3$. NMR spectroscopy was performed on a Bruker Avance NEO 600 MHz NMR spectrometer equipped with a TCI Prodigy CryoProbe (Bruker). Spectra were acquired at 298 K and consisted for each sample of a 1D $^1H$ PURGE spectrum[63], which was then phase corrected, baseline corrected and the chemical shifts were referenced to the TSP peak at 0.0 ppm.

Spectra were aligned by the PAFFT method (peak alignment by fast Fourier transform) and then normalized by probabilistic quotient normalization (PQN)[64]. To determine whether spectra cluster into groups, PLS-DA with k-fold cross-validation were performed. PLS-DA of the full 1D $^1H$ spectrum, after alignment, normalization using PQN, and log-scaling of the spectra was done. For multivariate analyses, spectra were scaled by the pareto method, which decreases the influence of high-intensity peaks while emphasizing low-intensity signals, such that both high and low-intensity signals have equal significance in the model[65]. $^1H$-NMR profiling of these cellular extracts allowed the identification of the main compounds present in the mixture. The metabolite assignment was done by comparing peak chemical shifts to those found in literature, in Human Metabolome Database (HMDB) and in the Biological Magnetic Resonance Bank (BMRB). Furthermore, all the metabolite structures were confirmed by 2D NMR experiments. Relative quantification of metabolites was achieved by integration of peaks to calculate peak area, which is proportional to the number of nuclei that give rise to the signal. Mean±SD of peak areas were calculated for each metabolite.

## Mass spectrometry

Cells were seeded in quintuplicate at concentration of 300 000 cells in a 6 well plate. The following day samples were treated with rotenone (0.5 μM) and antimycin A (0.5 μM) for 15 min or 1 h. Media was aspirated and cells were washed twice with 1 ml of ice-cold PBS. Metabolism was rapidly quenched by incubating samples on a dry-ice/methanol bath for 15 min. Cells from a parallel plate were counted to calculate the extraction volume to use for each sample. Samples were extracted with 1 ml extraction buffer/1 million cells (50:30:20 methanol:acetonitrile:water (LC-MS grade) with inclusion of 50 ng/mL HEPES), scraped on ice, transferred into eppendorf tubes and kept in agitation for 15 min at 4 °C on a rocker shaker. Then, cells were incubated for 1 h at −20 °C. Finally, samples were centrifuged twice for 10 min − 4 °C) at 16,000 g. Supernatants were transferred into autosampler vials and stored at −80 °C until analysis.

LC-MS analysis was performed using a Q Exactive Quadrupole-Orbitrap mass spectrometer coupled to a Vanquish UHPLC system (Thermo Fisher Scientific). The liquid chromatography system was fitted with a Sequant ZIC-pHILIC column (150 mm × 2.1 mm) and guard column (20 mm × 2.1 mm) from Merck Millipore (Germany) and temperature maintained at 35 °C.

The sample (3 μL) was separated at a flow rate of 0.1 mL/min. The mobile phase was composed of 10 mM ammonium bicarbonate and 0.15% ammonium hydroxide in water (solvent A), and acetonitrile (solvent B). A linear gradient was applied by increasing the concentration of A from 30 to 80% within 10 min and then maintained for 10 min.

The mass spectrometer was operated in positive polarity and with a SIM-targeted method for the molecular ions of AMP, ADP, ATP and adenosine (m/z 348.07036, 428.03669, 508.00302 and 268.10403 respectively) and resolution 70000. Major ESI source settings were: spray voltage 3.5 kv, capillary temperature 275 °C, sheath gas 35, auxiliary gas 5, AGC target 3e6, and maximum injection time 200 ms. The acquired spectra were analyzed using XCalibur Qual Browser and

XCalibur Quan Browser software (Thermo Scientific). Samples were analysed by quintuplicate.

## Metabolic assays

**ECAR and OCR measurement.** In 2D systems, oxygen consumption rate (OCR) and extracellular acidification rate (ECAR) were measured performing a Mitochondrial stress test (Cat. 103708-100) and a Glycolysis stress test (Cat. 103020-100) respectively, using a Seahorse XFe96 analyser and Wave Desktop and Controller 2.6 software (Agilent), following manufacturer instructions. Briefly, 5000 cells (15000 cells when comparing OCR from WM983A and WM983B) were seeded on Seahorse XF96 cell culture microplates in DMEM 10% FBS and incubated overnight at 37 °C and 10% $CO_2$. Next, media was changed to XF DMEM media, supplemented as indicated by the manufacturer and cells were incubated at 37 °C without $CO_2$ for 1 h, before proceeding immediately to perform the OCR or ECAR analysis. OCR was monitored following the sequential addition of oligomycin (1 μM), FCCP (0.25 μM) and rotenone/antimycin A (0.5 μM) and non-mitochondrial respiration was subtracted for the calculation of mitochondrial respiration. ECAR was monitored following the sequential addition of glucose (10 mM), oligomycin (1 μM) and 2-deoxy-D-glucose (2DG, 50 mM). OCR data were normalized by protein content where indicated.

For the measurement of OCR in 3D cell cultures, MitoXpress Xtra Oxygen Consumption assay (Cat. MX-200-4 Agilent) was used. This is a time resolved fluorescent assay that allows the measurement of oxygen consumption over time. Cellular respiration reduces extracellular oxygen levels, causing the MitoXpress Xtra reagent signal to increase, with the rate of increase (slope) reflecting the rate of oxygen consumption. Cells were embedded in a 3D collagen matrix as explained above. Twenty-four hours later, media was replaced with 60 μl DMEM 1% FBS containing MitoXpress Xtra reagent. Then, wells were sealed with HS mineral oil and time-resolved fluorescence (TR-F) was measured immediately for 2 h, using a BMG Omega plate reader and Omega 5.50 R4 software (BMG Labtech), following manufacturer instructions. Glucose oxidase (0.1 mg/ml) was used as a cell-free positive control. Antimycin A (1 μM) and FCCP (0.75 μM) were used as cell-based negative and positive control, respectively. Slope between 30 and 90 min was used to quantify OCR, following manufacturer's protocol, using Omega MARS 3.32 R5 software.

## Cellular ATP measurement

**Luminescent ATP detection assay.** For the measurement of total ATP, the luminescent ATP detection assay (Cat. ab113849 Abcam) was used. Briefly, the day before the assay 5000 cells/well were seeded on untreated and polyhema-treated (P3932, Sigma-Aldrich) 96 well plates and allow to grow overnight in DMEM 10% FBS, incubated at 37 °C and 10% $CO_2$. Cells cultured on polyhema plates were grown as suspension cells. The day after, ATP was quantified according to the manufacturer's protocol.

**Seahorse XF Real-Time ATP Rate Assay.** For the quantification of ATP production rate from both glycolytic and mitochondrial pathways, Agilent Seahorse XF Real-Time ATP Rate Assay was performed according to the manufacturer's instructions (Cat. 103592 Agilent Technologies) for the indicated conditions. For all adherent conditions, the day prior the assay 5000 cells/well, were seeded on a Seahorse XF96 cell culture microplate in DMEM 10% FBS, incubated at 37 °C and 10% $CO_2$. On the day of the experiment, media was changed for XF DMEM (pH=7.4) supplemented with 1 mM pyruvate, 2 mM glutamine and 10 mM glucose. For floating conditions, 7500 A375P cells, previously grown under floating conditions for 24 h, were seeded on the Seahorse XF96 cell culture microplate in the XF media specified above. Both adherent and floating cells were then incubated at 37 °C without $CO_2$ for 1 h. During this incubation period, oligomycin and rotenone/antimycin A were prepared in Seahorse media to achieve final concentrations of 1.5 μM and 0.5 μM respectively when injected. The glycolytic and mitochondrial ATP production rate were calculated according to manufacturer instructions and normalized by protein content using Wave Desktop and Controller 2.6 software. Data are shown as XF ATP rate index, which is indicative of the mitoATP Production Rate divided by glycoATP Production Rate.

**Measurement of ATP:ADP ratio in 3D.** PercevalHR (Addgene plasmid #49083) lentivirus were generated in HEK293T cells. Media with lentiviruses was added to recipient cell lines HT1080, A375M2 and A375P. Where indicated, PercevalHR expressing A375P cells were then transfected with the specified siRNA following the protocol as explained above. After 48 h cells were trypsinized, spun down and re-suspended in collagen I. Two hundred and fifty microliters of collagen-cell suspension were added per well in optical bottom 8-well μ-slide (Ibidi). Collagen was allowed to polymerize for 4 h at 37 °C and DMEM 10% FBS was added on top. Twenty-four hours later, media was replaced by FluoBrite DMEM for imaging. Perceval HR expressing cells were sequentially excited using 405 and 488 nm lasers with emission collected at 540 nm for both channels[21,66]. Z-stack images for the entire volume of the cell were taken using Zeiss LSM 880 with Fast Airyscan mode, Plan-Apochromat 63x/1.4 Oil DIC M27 objective lenses and Zen software (Carl Zeiss, Germany). Pixel-by-pixel ratio of ATP bound (488) / ADP bound (405) signals were calculated using ImageJ RatioPlus tool. Calibration bar shows minimal to maximal signal for each picture.

## Cell viability

Propidium iodide was used to stain dead cells. After spinning down spent media and cells, cells were resuspended in a solution of Propidium iodide (10 μg/ml) in PBS and incubated for 15 min on ice. Immediately afterwards, samples were analysed on a BD LSR Fortessa flow cytometer. Data were analysed using FlowJo 10.6 software.

## Immunohistochemistry

Paraffin-fixed sections were sequentially stained for DDR1, MFN2, OPA1, pMLC2, pAMPK and CD44, as previously described[15,67]. Tumour tissues were formalin-fixed and paraffin-embedded as per standard protocols. Sections of 4 μm thickness were heated at 60 °C for 1 h and then incubated in xylene and ethanol series, with $2 \times 5$ min $H_2O_2$/ethanol incubations to block endogenous peroxidase. Antigen retrieval was performed in Antigen Unmasking Solution pH 6 (H-3300, Vector Labs) using a pressure cooker system (110 °C for 10 min). Samples were washed in Dako Wash Buffer (S3006), before primary antibody incubation (40 min, 1:200 anti-DDR1, CST #5583), diluted in Antibody Diluent Reagent Solution (003218, invitrogen/ThermoFisher Scientific). Samples were washed and incubated with ImmPRESS® polymer secondary goat anti-rabbit antibody (goat anti-rabbit, RTU, Vector Labs, MP-7451) for 45 min. The reaction was developed using VIP peroxidase substrate solution (Vector Labs, SK-4600) for 10 min. All incubations were carried out at room temperature. Slides were counterstained with haematoxylin and mounted using DPX mounting medium (06522-500 ML, sigma). Slides were imaged using the Nano-Zoomer S210 slide scanner (Hamamatsu, Japan). The next day, slides were processed using the same procedure. Previous staining was stripped through the antigen retrieval step. In the second round, an anti-MFN2 primary antibody was used (40 min, 1:500, ab56889) and ImmPRESS® polymer secondary goat-anti mouse antibody (40 min, RTU, Vector Labs, MP-7452); developing, mounting and imaging was performed as before. The third round was performed as above, where slides were incubated with an anti-OPA1 antibody (40 min, 1:500, BD-612606) and ImmPRESS® polymer secondary goat-anti mouse antibody (40 min, RTU, Vector Labs, MP-7452) was used; developing, mounting and imaging was performed as before. In the forth round, anti- pMLC2 (ser19) antibody was added to the panel (40 min, 1:50, CST

#3671) and ImmPRESS® polymer secondary goat-anti rabbit antibody (40 min, RTU, Vector Labs, MP-7451) was used; developing, mounting and imaging was performed as before. In the following fifth round, pAMPK antibody (40 min, 1:150, CST#2535) and ImmPRESS® polymer secondary goat-anti-rabbit antibody (40 min, RTU, Vector Labs, MP-7451) were used; developing, mounting and imaging was performed as before. Finally, the sixth round included CD44 antibody to stain cell membranes which allowed the quantification of cell morphology. Tissue sections were incubated with anti-CD44 (40 min, 1:700, ab157107) and ImmPRESS® polymer secondary goat-anti-rabbit antibody (40 min, RTU, Vector Labs, MP-7451) was used; developing, mounting and imaging was performed as before. ImmPRESS® polymer secondary antibodies are ready to use.

**Image analysis.** Staining quantification was performed using QuPath 0.1.2 (Bankhead et al., 2017). For each marker, whole section images (WSI) from mouse tissue and TMAs were analysed performing positive cell detection, and three different thresholds were applied according to the intensity scores (0, 1, 2 and 3). Next, the software was trained by creating a random tree classification algorithm combined with the intensity information, in order to differentiate tumour from stroma, necrosis and immune cells. Then, IHC staining was graded semi-quantitatively by considering the percentage and intensity of the staining. A histologic score (Hs) was obtained from each sample, and values ranged from 0 (no immunoreaction) to 300 (maximum immunoreactivity) arbitrary units. The score was obtained by applying the following formula, Hs = 1 × (% light staining) + 2 × (% moderate staining) + 3 × (% strong staining). For co-localization analysis, images for the 5 markers (DDR1, MFN2, OPA1, pMLC2 and pAMPK) were aligned in FIJI v1.53t using TrackEM2 module. Next, colour deconvolution was performed using AEC-Haematoxylin vectors and a composite was created using channel-2 (red) for each staining[68]. The composite was adjusted inverting the LUT for each marker and given a pseudocolour. We used QuPath DoG superpixel segmentation to quantify cell morphology. CD44 staining was used to perform an accurate watershed segmentation. Finally, shape measurements were added in QuPath such as rounding in all the detections. To calculate the amoeboid score (As) the same ROI used in CD44 for each case we used with pMLC2 staining and the intensity for such marker was calculated in each detection. The As was calculated combining roundness and pMLC2 intensity for each detection, as shown in the formula: As=roundness (0-1) * Hs pMLC2 (0-300)= 0-300.

Values for As range from 0 to 300 were 0 represents a very elongated cell (roundness=0) with null levels of Myosin II (Hs=0), while the representative amoeboid cell harbours As of 300. For each case values are represented as mean of all detection per ROI.

**Gene enrichment analyses**
Normalized gene expression microarray data of rounded-amoeboid melanoma cells (GSE23764)[4] were analysed by comparing rounded-amoeboid A375M2 cells to more elongated and less contractile A375P cells or to A375M2 cells treated with ROCK1/2 inhibitors (H1152 and Y27632) or blebbistatin. Heatmaps were performed using MeV_4_9_0 software.

**Statistics and reproducibility**
Statistical analyses were performed using GraphPad Prism 9.5.0 software (GraphPad, San Diego). The following statistical analysis were used: one sample *t*-test, two-sided Student's *t*-test, paired *t*-test, Mann–Whitney's test, Wilcoxon test, two-sided one-way ANOVA with Tukey's, or Dunnett's post hoc test, Kruskal–Wallis with Dunn's multiple comparison test and two-way ANOVA with Dunnett's post hoc test. Data were represented in bar graphs as mean ± standard error of the mean (SEM), dot plots and violin plots as median with interquartile range and box plots as median (centre line), interquartile range (box)

and min-max values (whiskers). In general, experiments were carried out at least 3 independent times with 2-3 technical replicates, unless otherwise stated (indicated as n in figure legends). *P* values were calculated using two-tailed tests. *P* values of less than 0.05 were considered statistically significant.

## Data availability
Source data are provided with this paper. Gene expression dataset re-analysed in this study is available in NCBI GEO under accession number GSE23764. The Mass spectrometry data generated in this study have been deposited in the Metabo-Lights database under accession code MTBLS7755 (https://www.ebi.ac.uk/metabolights/MTBLS7755). All unique/stable reagents generated in this study are available with a completed Materials Transfer Agreement. Source data are provided with this paper.

## Code availability
The code generated during this study is available on GitHub [https://github.com/markholtuk/TFM][69].

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

## Acknowledgements

The work was supported by Cancer Research UK (CRUK) C33043/A12065 and C33043/A24478 (V.S.M., E.C.M., O.M., P.P.); Barts Charity (V.S.M., V.G., J.M., R.S., S.L.G.); Royal Society RG110591 (V.S.M.); Fundación Ramón Areces (E.C.M.); World Wide Cancer Research 22-0329 (V.S.M., V.G.); ISCIII/FEDER "Una manera de hacer Europa" FIS-PI1500711 and PI18/00573 (R.M.M.); CIBERONC CB16/12/0023 (R.M.M. and X.M.G.), and from the European Research Council (E.R.C.) under the European Union's Horizon 2020 research and innovation programme (StG- 851055 to A.E.A.). A.E.A. is supported by the Francis Crick Institute, which receives its core funding from Cancer Research UK (CC2214), the UK Medical Research Council (CC2214), and the Wellcome Trust (CC2214). We thank the Centre for Biomolecular Spectroscopy for access to biophysical infrastructure. The Centre was funded by the Welcome and British Heart Foundation grants to MRC (ref. 202767/Z/16/Z and IG/16/2/32273 respectively). We acknowledge the metabolic flux analysis facility of the Barts School of Medicine and Dentistry created with the support of the Barts and the London Charity - grant number MGU0401. We thank Kairbaan Hodivala-Dilke, Jeremy Carlton and Verónica Torrano for helpful discussions and Fredrik Wallberg for technical advice. Illustrations were created with BioRender.com.

## Author contributions

Conceptualization, V.S.M., E.C.M. Methodology, V.S.M., E.C.M., P.P., S.M., M.H., R.A.A., A.L.G., M.R.C., V.G., J.S., V.M., A.E.A. Investigation, E.C.M., O.M., P.P., S.M., J.M., A.L., G.C., M.H., A.L.G., R.A.A., V.G., J.S., R.S., S.L.G. Validation, E.C.M., O.M., P.P. Software, S.M., M.H. R.M.M. and X.M.G. provided human tissue samples; Writing – Original Draft, V.S.M., E.C.M. Writing – Review & Editing, V.S.M., E.C.M. Funding Acquisition, V.S.M., E.C.M. Resources, R.A.A., M.R.C., M.H.Supervision, V.S.M.

## Competing interests

The authors declare no competing interests.

## Additional information

[1]Barts Cancer Institute, Queen Mary University of London, John Vane Science Building, Charterhouse Square, London EC1M 6BQ, UK. [2]Randall Centre for Cell and Molecular Biophysics, New Hunt's House, Guy's Campus, King's College London, London SE1 1UL, UK. [3]Centre for Biomolecular Spectroscopy, King's College London, London SE1 1UL, UK. [4]Department of Dermatology, Hospital Universitari Arnau de Vilanova, University of Lleida, CIBERONC, IRB Lleida, Lleida 25198, Spain. [5]Department of Pathology and Molecular Genetics, Hospital Universitari Arnau de Vilanova, University of Lleida, IRB Lleida, CIBERONC, Lleida 25198, Spain. [6]Department of Pathology, Hospital Universitari de Bellvitge, University of Barcelona, IDIBELL, CIBERONC, L'Hospitalet de Llobregat, Barcelona 08907, Spain. [7]London Centre for Nanotechnology, University College London, London WC1H 0AH, UK. [8]Department of Cell and Developmental Biology, University College London, London WC1E 6BT, UK. [9]Cell and Tissue Mechanobiology Lab, The Francis Crick Institute, 1 Midland Road, London NW1 1AT, UK. [10]Department of Physics, King's College London, London WC2R 2LS, UK. [11]School of Cardiovascular and Metabolic Medicine & Sciences, King's College London BHF Centre of Research Excellence, London SE1 1UL, UK. [12]Present address: Institut de Pharmacologie et de Biologie Structurale (IPBS), UMR5089, CNRS-Université de Toulouse III-Paul Sabatier, BP 64182, 31077 Toulouse, Cedex 4, France. [13]These authors contributed equally: Vittoria Graziani, Oscar Maiques. ✉e-mail: v.sanz-moreno@qmul.ac.uk

