## [Peer Review File · Nature Communications]

REVIEWER COMMENTS

Reviewer #1 (Remarks to the Author):

The manuscript entitled "AMPK is a mechano-metabolic sensor linking cell adhesion and mitochondrial dynamics to Myosin II dependent cell migration" by Crosas-Molist et al. presents some impressive 3D traction measurements and interesting correlations that imply a mechano-metabolic relationship. However, in its current form the manuscript lacks a clear presentation of experimental evidence that connects the proposed mechanism. The manuscript could be improved with a streamlined focus on WM983A and A375P (lower AMP, elongated mitochondrial morphology, and matrix adhesions) paired with WM983B and A375M2 (higher AMP, fragmented mitochondria, amoeboid and metastatic). This would support the examination of AMPK, and AMPK's role of regulating mitochondrial fission via MFF. This finding would establish a reason to test if manipulations of AMPK or mitochondrial structure/function affect modes of migration, traction stress, and MYPT1 regulation.

Major comments

1. Currently, the respiration and ATP:ADP ratio data does little to benefit the story since ATP:AMP is what is sensed by AMPK and the respiration phenotype between the paired cell lines has a nominal difference. XF96 wells can accommodate far more cells than 5k, a larger cell number (~3x) is likely to aid in the comparative difference between the WM983A/WM983B and A375P/A375M2. Should a robust respiratory effect be observable between these cell line pairs, testing if mitochondrial perturbations (e.g., ETC inhibition or forced fragmentation) cause the AMP levels to rise in the WM983A and A375P with NMR-based metabolomics would be a more straightforward connection to AMPK.
2. Supplemental Figure 5 does present some excellent controls that indicate that compound C, A769662, and siPRKAA1/2 (AMPK knockdown) can be leveraged to articulate the mechanism proposed. For example, in Figure 5e, it would be more cohesive to show manipulations of AMPK activity to see what happens to branches or circularity (Figure 5g-h) when AMPK is active or inhibited. Similarly, this could also be applied to a data pertaining to cell migration/traction/morphology (e.g., Figure 2H-I, could be reproduced with AMPK or MFF manipulations in a later figure).
3. Is OCR (slope) the rate of change during repeated measures of the stress test treatments? Is there an established rationale or explanation for this assessment? Because at present the way these data are presented it remains unclear what information this provides towards the central thesis of this article.
4. The 3D traction force mapping methodology is impressive. It would be very convincing to see what happens to 3D traction stresses for a pair (mesenchymal/amoeboid) of cells with different migration modes treated with:
 - Non-hydrolysable ATP, demonstrating a requirement for ATP
 - Forced mitochondrial fragmentation (MFN KD) or branched network structure (MFF or DRP1 KD), respiratory poison, or mitochondrial ETC depletion (rho0)
 - Compound C, A769662, and siPRKAA1/2 (AMPK knockdown)
5. Overall the work presented in this manuscript is interesting, however the authors have included quite a lot of distracting data that fail to advance their central hypothesis. It is strongly recommended that the authors reduce the superfluous data and instead include fewer and more focused figures.

Minor points:

1. It is recommended that the author clarify "high levels" or "active" when describing AMPK.
2. With regard to Figures 2E-F, the Perceval HR could be assayed with a forced mitochondrial fragmentation or ETC poison to articulate the connection to mitochondria. As presented the background signal for A375M2 cell seems to be much higher. Please clarify. The dynamic range of Perceval HR is greater in cells cultured in 5 mM glucose media, FluoBrite DMEM appears to 25mM ("High glucose"), see Figure 3 in reference 21.
3. Pertaining to the DDR1 manipulations, it is unclear if these modifications alter the cell's mode of migration/3D traction stress commensurate with the expected bioenergetic profile (respiration/ATP:AMP)?
4. Regarding the bar graphs of OCR presented in Figure 2b, it appears that non-mitochondrial respiration was subtracted, which is the correct way to represent the data. Nevertheless this fact is not mentioned and should be included in the figure legend or method section so that this is clear to readers.

Reviewer #2 (Remarks to the Author):

In the paper entitled "AMPK is a mechano-metabolic sensor linking cell adhesion and mitochondrial dynamics to Myosin II-dependent cell migration" Crosas-Molist E. et al study the role of AMPK activation in low adhering migratory cells, which links mitochondrial dynamics and metabolism with cytoskeleton rearrangements.

They showed that the downregulation of DDR1, a collagen I receptor, is the main trigger of rounded-amoeboid migration, decreased OXPHOS, lower ATP levels, and AMPK activation that culminates with mitochondrial fragmentation.

Although the results are very exciting, this study is affected by some technical and conceptual problems that undermine the strength of the author's conclusions and that need to be addressed before publication.

Major points

1. The differences in mitochondrial metabolism across the different cell lines are unclear, mostly due to technical shortfalls. For instance, the Seahorse analysis should be normalized either by protein content or cell number. Indeed, the analysis in Supplementary Fig2F shows a difference between cell lines after rotenone and antimycin addition. This result is usually due to different cell numbers across conditions at the moment of the analyses. The normalisation to protein or cell number should reduce this effect. In addition, the authors need to clarify what OCR "slope" is and why they use this parameter instead of the more established basal OCR as a readout of mitochondrial respiration.

The authors should use better representative images for the TMRE experiments in Figure 5E as it seems there are no changes in mitochondrial activity, even though the quantification in Figure 5I looks clear. Moreover, the data obtained with Mitotracker Deep Red must be taken with caution as the staining with this dye depends on the mitochondrial potential. Ideally, the authors should assess the mitochondrial reticulum using the expression of a mitochondrially-targeted GFP or other mitochondrial-potential-independent probes, such as Mitotracker Green.

Finally, the authors need to assess whether the alterations of mitochondrial morphology performed across the study are associated with changes in metabolism, in particular changes in respiration and AMP levels, which are important players in their model.

The metabolomics data presented in Supplementary Figure 2B are unclear and they do not appear to support a mitochondrial defect in the amoeboid populations. For instance, most of the TCA cycle intermediates are elevated in the comparison between parental and metastatic cells. Overall, this analysis doesn't add much to the work unless it is properly used and/or supported by appropriate metabolic tracing. Of note, this referee doesn't believe that it is necessary to include this metabolomic work. The point here is that if it is included it has to be presented more appropriately.

2. The activation of AMPK in all the conditions tested needs to be confirmed by probing the phosphorylation of additional downstream targets, including pACC, and FAS.

Minor points

1- The western blot panels should include the size of the proteins (kDa), better labelling and should be wider. We recommend including the whole scans of the membranes.

2- The representation of the graphs in Figure 1D needs to be explained. Is this a representative trace?

3- In Figure 2G, second graph, it seems there are only two experiments performed. Authors need to perform at least 3 experiments to have statistical data. It is not clearly observed if two experiments are overlapping.

4- For statistical analysis, when fold changes are being compared (and control is 1) the one-sample t test analysis should be used (e.g Figure 3H, Supplementary Figure 5). In line with this, in some graphs it is not clear what error bars represent (e.g Figure 6D). In this case, the significance seems to

be very high, but the error bars are quite big.

5- The asterisk to indicate the statistical significance should be replaced by the actual p-value.

6- The use of the different cell lines is sometimes confusing. It would be ideal to include which cell line is being used in each panel, especially considering the characteristics of each of them (e.g Figure 4 M-O).

7- Some western blot panels are missing the loading control (Figure 5A, C; Supplementary Figure 6A, C).

8- Figure 9 must be referred in the discussion.

Reviewer #3 (Remarks to the Author):

In this study, Crosas-Molist et al. investigate cell migration mechanisms in 3D cancer lines, suggesting AMPK to be the key factor linking metabolic energy production and cytoskeleton dynamics in migrating cells. This link is very well demonstrated by the analysis of ATP production, cytoskeletal organization, adhesion levels and finally mitochondrial dynamics that together allow cancer cells invasion and dissemination.

The study is very precise and detailed, and the results are very well explained. Overall, this is a great and exciting study that support the rising concept of mechano-metabolic crosstalk.

My only minor comments are:

1. In discussion, it would be good to add the potential application of this discovery in cancer drug therapy, and metabolic related diseases.
2. As I understood ECAR/OCR were first measured in 2D system, and then confirmed in 3D. As the results were more significant and convincing in 3D, I fail to see why the 2D measurements are in the main figure. Perhaps this should also go to supplement. Also, the XF analyzer is a very sensitive machine that needs a lot of repetitions in order to quantify, however, the amount of "n" is not mentioned. In order to suggest that mitochondria are indeed the main energy production source, sufficient repetitions of the 2D-3D measurements are needed.

REVIEWER COMMENTS TO NCOMMS-22-19099

Reviewer #1 (Remarks to the Author):

The manuscript entitled “AMPK is a mechano-metabolic sensor linking cell adhesion and mitochondrial dynamics to Myosin II dependent cell migration” by Crosas-Molist et al. presents some impressive 3D traction measurements and interesting correlations that imply a mechano-metabolic relationship. However, in its current form the manuscript lacks a clear presentation of experimental evidence that connects the proposed mechanism. The manuscript could be improved with a streamlined focus on WM983A and A375P (lower AMP, elongated mitochondrial morphology, and matrix adhesions) paired with WM983B and A375M2 (higher AMP, fragmented mitochondria, amoeboid and metastatic). This would support the examination of AMPK, and AMPK’s role of regulating mitochondrial fission via MFF. This finding would establish a reason to test if manipulations of AMPK or mitochondrial structure/function affect modes of migration, traction stress, and MYPT1 regulation. We thank the reviewer for their positive and constructive feedback. We streamlined the manuscript by focussing in the comparison between the pairs of melanoma cell lines (WM983A/WM983B and A375P/A375M2) after the characterisation of the modes of migration and metabolism in the panel of cell lines representative of the elongated-mesenchymal and rounded-amoeboid mode of movement. We first describe the characteristics of individual cell migration modes (morphology, myosin II activity, adhesion and traction stress into the matrix), followed by the analysis of energy levels (ATP/ADP/AMP) that lead to differential AMPK activation. Finally, we link AMPK activation to both cytoskeleton (via MYTP1 phosphorylation) and mitochondrial (via MFF phosphorylation) remodelling, leading to the regulation of cells’ mode of migration.

Major comments

1. Currently, the respiration and ATP:ADP ratio data does little to benefit the story since ATP:AMP is what is sensed by AMPK and the respiration phenotype between the paired cell lines has a nominal difference. XF96 wells can accommodate far more cells than 5k, a larger cell number (~3x) is likely to aid in the comparative difference between the WM983A/WM983B and A375P/A375M2. Should a robust respiratory effect be observable between these cell line pairs, testing if mitochondrial perturbations (e.g., ETC inhibition or forced fragmentation) cause the AMP levels to rise in the WM983A and A375P with NMR-based metabolomics would be a more straightforward connection to AMPK.

We agree with the reviewer that AMP is the main activator of AMPK. Following the reviewer’s suggestions, to improve the connection to AMPK, we have now analysed and added the levels of AMP and ratios ATP/AMP and ATP/ADP upon mitochondrial electron chain inhibition (ETC) in Fig. 4a, b and Supplementary Fig. 5a. We treated elongated-mesenchymal A375P and WM983A cells with Rotenone and Antimycin A for 15 minutes and 1 hour and analysed the levels of ATP, ADP and AMP. As expected, results showed an increase in AMP and a decrease in ATP/AMP and ATP/ADP ratios upon ETC inhibition that led to an increase in AMPK phosphorylation and activation (Fig 4a, b and Supplementary Fig. 5a).

On the other hand, an imbalance in ATP/ADP also leads to AMPK phosphorylation and activation (doi.org/10.1038/nrm3311, doi: 10.1038/nature09932). Therefore, we consider informative to show the ATP/ADP ratio for a complete overview of the energy differences between the conditions compared. We believe that old and new data shown in 3D collagen I matrix using the PercevalHR biosensor measuring ATP/ADP ratios (Fig. 2c, d; Fig. 3j; Fig. 7d) support the whole data set.

This project aims to compare different modes of **individual cell migration** (not groups of cells that establish many cell-cell contacts). For that reason, we work at low cell confluence conditions. We thank the reviewer for the suggestion of increasing the cell number for the Seahorse analyses when comparing the pairs of cell lines. We found advantageous to increase the cell number to 15 K cells/well when

comparing the WM983 pair and we added this new set of data in Supplementary Fig. 2e-g. However, we found that 5 K cells/well was better to compare the A375 pair of cell lines since cells were over confluent and formed clumps when seeded at 15 K cells/well. For the reviewer, pictures of the 2 cell line pairs when seeded at 5K and 15K cells/well showing that 15K cells/well and 5K cells/well are appropriate for comparing the WM983 and A375 pairs, respectively (Fig. R1.1a). Also, graphs showing Oxygen consumption rates in the original manuscript vs the revised version that normalises for protein (Fig. R1.1b, c).

Fig. R1.1: a) Bright field images of the indicated cell lines at the indicated cell densities, taken from the Seahorse plate prior running the Seahorse experiment. b) Data from the original manuscript (Fig. 2b) showing oxygen consumption rate (OCR) in a panel of cell lines (5K cells/well) (n=3). c) New data showing OCR comparing WM983A vs WM983B (15K cells/well) and A375P vs A375M2 (5K cells/well) and normalized by protein content (n=4). Graphs show mean with SEM and p value by paired t-test.

2. Supplemental Figure 5 does present some excellent controls that indicate that compound C, A769662, and siPRKAA1/2 (AMPK knockdown) can be leveraged to articulate the mechanism proposed.

For example, in Figure 5e, it would be more cohesive to show manipulations of AMPK activity to see what happens to branches or circularity (Figure 5g-h) when AMPK is active or inhibited.

Similarly, this could also be applied to a data pertaining to cell migration/traction/morphology (e.g., Figure 2H-I, could be reproduced with AMPK or MFF manipulations in a later figure).

We agree with the reviewer on the relevance of showing that manipulations of AMPK activity alters mitochondrial morphology. These data were already included in the original manuscript in Fig. 5a-d and Supplementary Fig. 6a-c (figures remain the same in the current version). These data already showed that activation of AMPK with A769662 in WM983A and A375P induces mitochondrial fragmentation (decrease in the number of mitochondrial branches) while AMPK knockdown (via siPRKAA1/2) in WM983B and A375M2 cells resulted in mitochondrial fusion (increase in the number of mitochondrial branches). We also showed in previous version that manipulations of AMPK (inhibition via Compound C or siPRKAA1/2 in WM983B and A375M2, and activation via A769662 in WM983A and A375P) resulted in changes in the mode of migration (affecting cell morphology and

myosin II activity), originally in Fig. 4c-j and Supplementary Fig. 5 and currently in Fig. 4 d-l and Supplementary Fig. 5c-m. In addition, in this revised manuscript we show that reducing AMPK expression by RNAi (siPRKAA1/2) in A375M2 cells results in loss of amoeboid features accompanied by an increase in traction stress exerted by cells in the collagen matrix (Fig. 4h).

In the original manuscript, Fig. 6a-f showed that MFF knockdown in A375M2 cells resulted in loss of amoeboid features and decrease in cell invasion. Now in new Fig. 7e-i and Supplementary Fig. 7g-k (originally Fig. 6j-n and Supplementary Fig. 7d-i) we also showed that knock-down of MFN1/2 in elongated-mesenchymal cells (WM983A and A375P) induces a transition to rounded-amoeboid migration (cell rounding, increase of myosin II activity, decrease in adhesion and increased invasion). In the revised manuscript, we now show that upon MFN1/2 knockdown we can also measure a decrease in traction stresses into the matrix (Fig. 7j). Finally, we had originally shown that transition to amoeboid migration upon MFN1/2 knock-down is dependent on AMPK activation since it can be prevented with the AMPK inhibitor Compound C (Fig. 6o-s in the original manuscript and Figure 7k-o in revised version).

Finally, in the revised manuscript, we have also added XF ATP rate index, indicative of the ratio between mitochondrial and glycolytic ATP in A375M2 cells. New Supplementary Fig. 5h and Supplementary Fig. 7c show that there is an increase in the XF ATP rate index when AMPK or MFF expression is reduced, indicative of an increase in mitochondrial derived ATP.

3. Is OCR (slope) the rate of change during repeated measures of the stress test treatments? Is there an established rationale or explanation for this assessment? Because at present the way these data are presented it remains unclear what information this provides towards the central thesis of this article.

Individual cells can fully adopt elongated-mesenchymal or rounded-amoeboid modes of migration when they are migrating in a pliable 3D environment. Some of the features are conserved when cells are seeded in 2D (regular cell culture plates). Nevertheless, it is when cells are grown in a 3D pliable environment that we are able to detect certain properties such as cytoskeletal remodelling and larger differences in cell metabolism.

In order to measure mitochondrial respiration in 3D, we used MitoXpress Xtra Oxygen Consumption Assay (Agilent) in Fig. 2a, b and 7c of the revised manuscript. OCR (slope) is indicative of the oxygen consumption rate in cells embedded in a 3D collagen I matrix. This time resolved fluorescent assay allows the measurement of oxygen consumption over time. Cellular respiration reduces extracellular oxygen levels, causing the MitoXpress Xtra reagent signal to increase, with the rate of increase (slope) reflecting the rate of oxygen consumption, the higher the slope, the faster is the oxygen being consumed, the more the cells engage in mitochondrial respiration (doi.org/10.1089/zeb.2020.1878). This is now better explained in materials and methods (page 32).

4. The 3D traction force mapping methodology is impressive. It would be very convincing to see what happens to 3D traction stresses for a pair (mesenchymal/amoeboid) of cells with different migration modes treated with:

- Non-hydrolysable ATP, demonstrating a requirement for ATP
- Forced mitochondrial fragmentation (MFN KD) or branched network structure (MFF or DRP1 KD), respiratory poison, or mitochondrial ETC depletion (rho0)
- Compound C, A769662, and siPRKAA1/2 (AMPK knockdown)

As suggested by reviewer, we have added traction stress measurements in:

1. Elongated-mesenchymal A375P cells after forced mitochondrial fragmentation (siMFN1/2) (Fig. 7j)
2. Rounded-amoeboid A375M2 cells after reducing AMPK expression (Fig. 4h).

Results showed lower traction stress applied by cells adopting rounded-amoeboid features (A375P siMFN1/2 and A375M2 Scramble) when compared to cells harbouring elongated-mesenchymal characteristics (A375P Scramble and A375M2 siAMPK), in agreement with the data shown in Fig.1.

Cell migration requires energy due to actomyosin-based activity and actin polymerization occurring during cell movement (doi.org/10.1242/jcs.248385; doi.org/10.1016/j.tcb.2022.09.009). Myosin II is the major motor protein that regulates actomyosin contractility and converts ATP into mechanical force. The ATPase function of the Myosin II regulatory light chains enables the translocation of the actin filaments towards their barbed ends (doi.org/10.1016/j.tcb.2007.02.002). Cells treated with contractility/migration inhibitors lower ATP hydrolysis (doi.org/10.1091/mbc.E17-01-0041), and when migrating in complex 3-dimensional (3D) environments, cells choose the path with minimal energy cost (doi.org/10.1038/s41467-019-12155-z). In our manuscript, we show that treatment with blebbistatin, a selective inhibitor of myosin II ATPase activity (doi.org/10.1007/s10974-004-6060-7), completely abolishes traction stress in both elongated-mesenchymal and rounded-amoeboid cells (Fig. 1i). We believe that these results support the requirement of ATP during any mode of cell migration. We used blebbistatin as a more specific approach to study cell migration than using non-hydrolysable ATP because the latter would interfere not only with cell migration, but with other high-energy demanding cellular processes such as macromolecular biosynthesis.

5. Overall the work presented in this manuscript is interesting, however the authors have included quite a lot of distracting data that fail to advance their central hypothesis. It is strongly recommended that the authors reduce the superfluous data and instead include fewer and more focused figures.

We have now removed non-essential data and re-structured some figures to better guide the reader through the key messages of the manuscript.

- In Fig. 2 we now show data regarding mitochondrial respiration in a 3D collagen I matrix while we have moved mitochondrial OCR and ATP levels measured in 2D (from Seahorse) to Supplementary Fig. 2. We have also moved the metabolomics PLS-DA analysis to Supplementary Fig. 2 and we have removed the NMR spectra and the heatmap of the metabolites that seemed distracting, as also suggested by reviewer 2.
- Fig. 5 shows detailed comparison of the 2 pairs of cell lines (WM983 and A375), while data from HT1080 and MDA-MB-231 have been moved to Supplementary Fig. 6.
- Fig. 6 is now split into 2 figures: New Fig. 6 shows the effects of impairing mitochondrial fission in rounded-amoeboid cells while new Fig. 7 shows the effects of impairing mitochondrial fusion in elongated-mesenchymal cells.
- Also in new Fig. 7 and Supplementary Fig. 7, only data from the MFN1/2 double knockdown in elongated-mesenchymal cells are shown and data from the singles knockdowns (MFN1 and MFN2) have been removed altogether from the manuscript.

We believe that these changes improved the focus of the figures and the flow of the manuscript.

Minor points:

1. It is recommended that the author clarify “high levels” or “active” when describing AMPK.

We have now explained at the start of the manuscript how AMPK activation is inferred by phosphorylation levels of AMPK (pThr172-AMPK). Thereafter, we specify either high levels of phosphorylated AMPK (pAMPK) or AMPK activation. Moreover, we now show AMPK downstream kinase substrate pSer79ACC/ACC throughout the manuscript.

2. With regard to Figures 2E-F, the Perceval HR could be assayed with a forced mitochondrial fragmentation or ETC poison to articulate the connection to mitochondria. As presented the background signal for A375M2 cell seems to be much higher. Please clarify. The dynamic range of Perceval HR is greater in cells cultured in 5 mM glucose media, FluoBrite DMEM appears to 25mM (“High glucose”), see Figure 3 in reference 21.

As suggested by reviewer, we have now analysed the ATP/ADP ratio using PercevalHR in A375P cells after reducing MFN1/2 levels. These data are now in Fig. 7d. Results showed a decrease in ATP/ADP after MFN1/2 siRNA transfection. These data together with the increase in AMP and the decrease in ATP/AXP ratios upon ETC inhibition with Rotenone and Antimycin A in A375P cells (shown in Fig. 4a), demonstrate a clear connection between mitochondrial morphology, mitochondrial activity and ATP/ADP/AMP intracellular levels.

Colours shown in the pictures from PercevalHR analysis are related to the calibration bar shown in each picture, which is automatically determined by the ‘ratio plus’ plug in from the software Fiji and that we are unable to modify. Black is given to the minimal and white to the maximal value in the image. In Fig. 2d (Fig. 2c in the revised manuscript), for HT1080 cells the lowest value is 1 (meaning same signal coming from ATP and ADP) and corresponds to the area with no cells. Levels of ATP are higher than levels of ADP throughout the HT1080 cell. However, for A375M2 cells, the lowest value is 0.99, meaning that some pixels within the cell have a greater signal for ADP than for ATP, while the areas with no cells show a value of 1 (same signal from ATP and ADP), which correspond to blue in the calibration bar. Therefore, the blue shown in the area with no cells for A375M2 is equivalent to the black in the area with no cells for HT1080 and corresponds to a ratio ATP/ATP of 1. We have now explained this better in materials and methods in page 34.

Following the reviewer’s suggestion, we repeated the comparison between HT1080 and A375M2 cells from Fig. 2d, e (Fig. 2c, d in the revised manuscript) using 5mM and 25mM glucose DMEM and we obtained very similar data in both conditions (for the reviewer Fig. R1.2). Given these data and the advantage of using Fluorobrite for a better signal to noise ratio, we performed the remaining experiments using Fluorobrite media, which is only available at 25mM Glucose.

Fig. R1.2: Quantification of ATP:ADP ratio in HT1080 and A375M2 cells embedded in a collagen I matrix, cultured with DMEM 5mM Glucose or 25mM Glucose (>12 cells/condition pooled from 3 experiments). Box plots show min to max and p value was calculated with Kruskal-Wallis with Dunn’s multiple comparisons test.

3. Pertaining to the DDR1 manipulations, it is unclear if these modifications alter the cell's mode of migration/3D traction stress commensurate with the expected bioenergetic profile (respiration/ATP:AMP)?

Reducing DDR1 expression levels in A375P cells induced cytoskeletal remodelling and a clear transition to a rounded-amoeboid mode of migration. In the original version of the manuscript, we showed in Fig. 3i, j (same figure in the revised manuscript) that this transition was accompanied with a decrease in mitochondrial respiration and ATP/ADP in a 3D collagen I matrix . Moreover, in Fig. 4k-o (Fig. 4m-q in the revised manuscript) we observed an increase in phospho-AMPK as a result of reducing DDR1 levels, due to the ATP/ADP imbalance, and this is essential for the acquisition of amoeboid features. AMPK is crucial for the process since the transition was prevented when cells were treated with AMPK inhibitor Compound C (Fig. 4k-o (Fig. 4m-q in the revised manuscript)). Altogether, these data indicate that after reducing DDR1 expression, there is an energy imbalance that activates AMPK pathway promoting rounded-amoeboid migration.

4. Regarding the bar graphs of OCR presented in Figure 2b, it appears that non-mitochondrial respiration was subtracted, which is the correct way to represent the data. Nevertheless this fact is not mentioned and should be included in the figure legend or method section so that this is clear to readers. A sentence has been added in the methods section clarifying that non-mitochondrial respiration was subtracted when calculating mitochondrial respiration (page 32).

Reviewer #2 (Remarks to the Author):

In the paper entitled “AMPK is a mechano-metabolic sensor linking cell adhesion and mitochondrial dynamics to Myosin II-dependent cell migration” Crosas-Molist E. et al study the role of AMPK activation in low adhering migratory cells, which links mitochondrial dynamics and metabolism with cytoskeleton rearrangements.

They showed that the downregulation of DDR1, a collagen I receptor, is the main trigger of rounded-amoeboid migration, decreased OXPHOS, lower ATP levels, and AMPK activation that culminates with mitochondrial fragmentation.

Although the results are very exciting, this study is affected by some technical and conceptual problems that undermine the strength of the author’s conclusions and that need to be addressed before publication.

We thank the reviewer for their positive and constructive feedback.

Major points

1. The differences in mitochondrial metabolism across the different cell lines are unclear, mostly due to technical shortfalls. For instance, the Seahorse analysis should be normalized either by protein content or cell number. Indeed, the analysis in Supplementary Fig2F shows a difference between cell lines after rotenone and antimycin addition. This result is usually due to different cell numbers across conditions at the moment of the analyses. The normalisation to protein or cell number should reduce this effect. In addition, the authors need to clarify what OCR “slope” is and why they use this parameter instead of the more established basal OCR as a readout of mitochondrial respiration.

We show now mitochondrial respiration and mitochondrial ATP comparing the two pairs of cell lines (WM983A/B and A375P/M2) normalized by protein content in Supplementary Fig. 2e-g. After normalisation we still find a decrease in mitochondrial respiration in cells adopting a rounded-amoeboid mode of migration when compared to their elongated-mesenchymal pairs.

Individual cells can fully adopt elongated-mesenchymal or rounded-amoeboid modes of migration when they are in a pliable 3D environment. Some of the features are conserved when cells are seeded in 2D (regular cell culture plates). Nevertheless, it is when they are grown in a 3D pliable environment that we are able to detect certain properties such as cytoskeletal remodelling and larger differences in cell metabolism.

In order to measure mitochondrial respiration in 3D, we used MitoXpress Xtra Oxygen Consumption Assay (Agilent) in Fig. 2a, b and 7c of the revised manuscript. OCR (slope) is indicative of the oxygen consumption rate in cells embedded in a 3D collagen I matrix. This time resolved fluorescent assay allows the measurement of oxygen consumption over time. Cellular respiration reduces extracellular oxygen levels, causing the MitoXpress Xtra reagent signal to increase, with the rate of increase (slope) reflecting the rate of oxygen consumption, the higher the slope, the faster is the oxygen being consumed, the more the cells engage in mitochondrial respiration (doi.org/10.1089/zeb.2020.1878). This is now better explained in materials and methods (page 32).

The authors should use better representative images for the TMRE experiments in Figure 5E as it seems there are no changes in mitochondrial activity, even though the quantification in Figure 5I looks clear. Moreover, the data obtained with mitotracker deep red must be taken with caution as the staining with this dye depends on the mitochondrial potential. Ideally, the authors should assess the mitochondrial reticulum using the expression of a mitochondrially-targeted GFP or other mitochondrial-potential-independent probes, such as mitotracker green.

We apologize for the lack of clarity in the method of measuring mitochondrial membrane potential using TMRE. We used Mitotracker Deep red in order to visualize mitochondria in cells stably expressing LifeAct-GFP, which allowed us to analyse the crosstalk between mitochondrial features and cytoskeleton. However, Mitotracker Deep red intensity was not taken into account when analysing the mitochondrial membrane potential, but the staining was only used to generate a mask for mitochondria, to analyse TMRE intensity only in those areas corresponding to mitochondria. This is now better explained in materials and methods (pages 28-29).

Nevertheless, to ensure that effects of mitochondrial membrane potential on Mitotracker Deep red were not altering the measurements for TMRE intensity, these experiments were repeated using Mitotracker Green and TMRE in naïve cells (not expressing LifeAct-GFP). New data (pictures and quantification) comparing the two pairs of melanoma cell lines (WM983A/B and A375P/M2) are now shown in Fig. 5f. This new dataset confirms lower mitochondrial membrane potential in rounded-amoeboid cells compared to their elongated-mesenchymal counterparts.

Finally, the authors need to assess whether the alterations of mitochondrial morphology performed across the study are associated with changes in metabolism, in particular changes in respiration and AMP levels, which are important players in their model.

We have now added new data analysing ATP/ADP ratio using PercevalHR (Fig. 7d) and mitochondrial respiration in 3D (Fig. 7c) and 2D environments (Supplementary Fig. 7f), in A375P cells after manipulating MFN1/2. These data indicated that reducing mitochondrial fusion resulted in a decrease in mitochondrial respiration, especially in a 3D environment, that was associated with a decrease in ATP/ADP ratio, leading to AMPK phosphorylation and subsequent transition to a rounded-amoeboid mode of migration (Fig. 7e-o). We have also included new data showing that reducing MFF expression levels in A375M2 cells induced an increase in mitochondrial respiration (Supplementary Fig. 7b), associated to an increase in XF ATP rate (mitochondrial/glycolytic ATP ratio) (Supplementary Fig. 7c). All of these events were associated with decreased AMPK phosphorylation and loss of amoeboid features (Fig. 6).

An imbalance in ATP/ADP leads to AMPK phosphorylation and activation (doi.org/10.1038/nrm3311, [doi: 10.1038/nature09932](https://doi.org/10.1038/nature09932)), and can be measured in a 3D environment by using the PercevalHR biosensor. Therefore, we consider informative to show the ATP/ADP ratio for a complete overview of the energy differences between the conditions compared. We believe that old and new data shown in 3D collagen I matrix using the PercevalHR biosensor measuring ATP/ADP ratios (Fig. 2c, d; Fig. 3j; Fig. 7d) supports the whole data set.

Additionally, following the reviewer 1's suggestion, we have now analysed and added the levels of AMP and ratios ATP/AMP and ATP/ADP upon mitochondrial electron chain inhibition (ETC) in Fig. 4a, b and Supplementary Fig. 5a. We treated elongated-mesenchymal A375P and WM983A cells with Rotenone and Antimycin A for 15 minutes and 1 hour and analysed the levels of ATP, ADP and AMP. As expected, results showed an increase in AMP and a decrease in ATP/AMP and ATP/ADP ratios upon ETC inhibition that led to an increase in AMPK phosphorylation and activation (Fig 4a, b and Supplementary Fig. 5a). This is in agreement with previous work from Toyama et al reporting that ETC poisons induce mitochondrial fission via AMPK activation and MFF phosphorylation ([doi: 10.1126/science.aab4138](https://doi.org/10.1126/science.aab4138)), linking mitochondrial morphology to energy levels and AMPK activation.

The metabolomics data presented in Supplementary Figure 2B are unclear and they do not appear to support a mitochondrial defect in the amoeboid populations. For instance, most of the TCA cycle intermediates are elevated in the comparison between parental and metastatic cells. Overall, this analysis doesn't add much to the work unless it is properly used and/or supported by appropriate

metabolic tracing. Of note, this referee doesn't believe that it is necessary to include this metabolomic work. The point here is that if it is included it has to be presented more appropriately.

We thank the reviewer for the advice. We have now removed the spectra and the heatmap in Supplementary Fig. 2a, b and moved the PLS-DA analysis to Supplementary Fig. 2a, just to show that the metabolome of elongated-mesenchymal and rounded-amoeboid cells is statistically different.

2. The activation of AMPK in all the conditions tested needs to be confirmed by probing the phosphorylation of additional downstream targets, including pACC, and FAS.

pACC/ACC immunoblots and their quantifications have been added throughout the manuscript confirming the activation or inhibition of the AMPK pathway:

- Fig. 4b, c, d, i, m, n
- Supplementary Fig. 5b, c, d, i, j
- Fig. 5a, c
- Supplementary Fig. 6a, c
- Fig. 6a
- Fig. 7k, l
- Supplementary Fig. 7a

We didn't observe changes in FAS for the different conditions (Fig. R2.1). Rounded-amoeboid migration is characterised by a fast turnover of bleb structures in the plasma membrane and the lipid composition of the plasma membrane may play a crucial role. We have some preliminary data suggesting that lipid metabolism plays a role in rounded-amoeboid migration. We believe that FAS expression might be regulated not only by AMPK activation but also by other factors and deserves a further analysis.

Figure 4

Figure 5

Figure 6

Figure 7

Supplementary Figure 5

Supplementary Figure 6

Supplementary Figure 7

Fig. R2.1: Representative western blot of FAS for the different figure panels where AMPK and ACC phosphorylation was regulated.

Minor points

1- The western blot panels should include the size of the proteins (KDa), better labelling and should be wider. We recommend including the whole scans of the membranes.

We added the molecular weight in all western blot panels and submitted a file with the whole scans of the membranes as per Nature instructions.

2- The representation of the graphs in Figure 1D needs to be explained. Is this a representative trace?

Graphs in Figure 1D correspond to a representative fluorescence intensity line scan and are explained in the figure legend:

‘Representative fluorescence intensity line scans (dashed white lines in image) showing distribution of F-actin (red), pMLC2 (green) and nucleus (blue) along elongated-mesenchymal (HT1080 and A375P) and rounded-amoeboid (A375M2) cells (n=10 cells/cell line).’

3- In Figure 2G, second graph, it seems there are only two experiments performed. Authors need to perform at least 3 experiments to have statistical data. It is not clearly observed if two experiments are overlapping.

We confirm that there are 3 experiments in Fig. 2g, two of them resulted in very similar data for the A375 pair and that is why points overlap (see below the 3 independent experiments for the 2 pair of cell lines).

Fig. R2.2: AMP levels quantified by $^1\text{H-NMR}$ for the indicated cell lines (n=3). Each dot represents a biologically independent experiment. p value by paired t-test.

4- For statistical analysis, when fold changes are being compared (and control is 1) the one-sample t test analysis should be used (e.g Figure 3H, Supplementary Figure 5). In line with this, in some graphs it is not clear what error bars represent (e.g Figure 6D). In this case, the significance seems to be very high, but the error bars are quite big.

We thank the reviewer for the suggestion. We have now modified the statistical analysis to one-sample t-test analysis when fold changes are compared:

- Fig. 3h
- Supplementary Fig. 5b, d, j
- Fig. 7i
- Supplementary Fig. 7k
- Supplementary Fig. 8g

We also ensured that all error bars are always explained in figure legend. In general, data are represented in bar graphs as mean \pm standard error of the mean (SEM), dot plots or violin plots as median with

interquartile range and box plots as median (centre line), interquartile range (box) and min-max values (whiskers).

In Fig. 6d in particular, data were represented in dot plot, violin plot in the revised manuscript, with median with interquartile range and a Kruskal-Wallis test with Dunn's multiple comparisons test was applied since data don't follow a normal distribution.

5- The asterisk to indicate the statistical significance should be replaced by the actual p-value.

Asterisks have now been replaced by the p-value.

6- The use of the different cell lines is sometimes confusing. It would be ideal to include which cell line is being used in each panel, especially considering the characteristics of each of them (e.g Figure 4 M-O).

We added the name of the cell line used in each panel.

7- Some western blot panels are missing the loading control (Figure 5A, C; Supplementary Figure 6A, C).

We apologise for that. We have added loading controls to all western blot panels.

8- Figure 9 must be referred in the discussion.

Figure 9 (Fig. 10 in the revised manuscript) is now cited in the discussion.

Reviewer #3 (Remarks to the Author):

In this study, Crosas-Molist et al. investigate cell migration mechanisms in 3D cancer lines, suggesting AMPK to be the key factor linking metabolic energy production and cytoskeleton dynamics in migrating cells. This link is very well demonstrated by the analysis of ATP production, cytoskeletal organization, adhesion levels and finally mitochondrial dynamics that together allow cancer cells invasion and dissemination.

The study is very precise and detailed, and the results are very well explained. Overall, this is a great and exciting study that support the rising concept of mechano-metabolic crosstalk.

We thank the reviewer for their positive and constructive feedback.

My only minor comments are:

1. In discussion, it would be good to add the potential application of this discovery in cancer drug therapy, and metabolic related diseases.

This has now been added to the discussion.

‘Understanding the mechanisms underlying cell migration plasticity is essential to unravel new vulnerabilities of cancer metastasis with therapeutic potential. AMPK dependence might therefore be an Achilles heel in highly-metastatic rounded-amoeboid cells. However, AMPK plays different roles in tumorigenesis, enabling metabolic adaptation under specific stress conditions such as hypoxia or glucose deprivation, favouring cancer cells survival but also suppressing cancer cell proliferation and tumour formation mainly via mTOR regulation⁵⁶. Further studies are required to clarify when and how AMPK antagonists will be beneficial to prevent cancer dissemination. In addition, our data suggest that the usage of compounds that directly or indirectly activate AMPK in other pathologies such as diabetes, mitochondrial disease and cardiovascular diseases⁵⁷, might need to be carefully evaluated, particularly in those patients with advanced cancer.’

2. As I understood ECAR/OCR were first measured in 2D system, and then confirmed in 3D. As the results were more significant and convincing in 3D, I fail to see why the 2D measurements are in the main figure. Perhaps this should also go to supplement. Also, the XF analyzer is a very sensitive machine that needs a lot of repetitions in order to quantify, however, the amount of “n” is not mentioned. In order to suggest that mitochondria are indeed the main energy production source, sufficient repetitions of the 2D-3D measurements are needed.

We agree with the reviewer and OCR data measured in 2D have been moved to Supplementary Fig. 2e-g. We also ensured that the “n” is indicated in every figure legend. Supplementary Fig. 2b, c n=3 (ECAR) and Supplementary Fig. 2e-g n=4 (OCR).

REVIEWERS' COMMENTS

Reviewer #1 (Remarks to the Author):

I am satisfied with the revised manuscript which addresses each of the critiques raised by all three reviewers. The authors have done a commendable job improving the article including adding additional experimental data.

Reviewer #2 (Remarks to the Author):

The authors did an excellent job in addressing the experimental concerns raised previously. The conclusions of the manuscript have been strengthened, and the work is now ready for publication. Congratulations on the extensive work!

There are some minor requests about the new Seahorse data. We would suggest the authors to indicate on the respective graphs when the experiment was done in 3D versus 2D to improve clarity. Also, although in the supplementary 2E-G they do add the normalization to protein (OCR), when calculating the "slope" using (Fig2A-B) it is unclear whether the data were normalised to protein. Finally, it should be clarified whether the ECAR measurements (FigSuppl 2B-C) were normalised for proteins.

Reviewer #3 (Remarks to the Author):

The authors have addressed all of our concerns, and the manuscript is now suitable for publication.